# Phytoextracts for Human Health from Raw and Roasted Hazelnuts and from Hazelnut Skin and Oil: A Narrative Review

**DOI:** 10.3390/nu15112421

**Published:** 2023-05-23

**Authors:** Mariangela Rondanelli, Mara Nichetti, Valentina Martin, Gaetan Claude Barrile, Antonella Riva, Giovanna Petrangolini, Clara Gasparri, Simone Perna, Attilio Giacosa

**Affiliations:** 1IRCCS Mondino Foundation, 27100 Pavia, Italy; mariangela.rondanelli@unipv.it; 2Unit of Human and Clinical Nutrition, Department of Public Health, Experimental and Forensic Medicine, University of Pavia, 27100 Pavia, Italy; 3Endocrinology and Nutrition Unit, Azienda di Servizi Alla Persona ‘‘Istituto Santa Margherita’’, University of Pavia, 27100 Pavia, Italy; dietista.mara.nichetti@gmail.com (M.N.); valentina.martin85@gmail.com (V.M.); gaetanclaude.barrile01@universitadipavia.it (G.C.B.); 4Development Department, Indena SpA, 20139 Milan, Italy; antonella.riva@indena.com (A.R.); giovanna.petrangolini@indena.com (G.P.); 5Division of Human Nutrition, Department of Food, Environmental and Nutritional Sciences (DeFENS), Università degli Studi di Milano, 20133 Milan, Italy; simoneperna@hotmail.it; 6CDI (Centro Diagnostico Italiano), 20147 Milan, Italy; attilio.giacosa@gmail.com

**Keywords:** hazelnut, hazelnut skin, hazelnut oil, hazelnut kernel

## Abstract

The objectives of this narrative review are as follows: an evaluation of the bromatological composition of hazelnuts and a comparison of the nutritional properties of raw versus roasted hazelnuts, taking into account potential differences among varieties from different production territories such as Turkey, Italy, Chile, and New Zealand; an evaluation of nutrients contained in hazelnut skin; and an evaluation of nutrients contained in hazelnut oil. This review incorporates 27 scientific articles that measured and reported the concentrations of macro- and micro-nutrients in hazelnuts. These hazelnuts were subjected to different processing methods, originated from various geographical areas, or belonged to different varieties. Our results showed that the different varieties and territories where the hazelnuts were cultivated influence their bromatological composition, and we found that different processing steps can largely influence the concentration of specific nutrients. The removal of the skin, which contains a very high concentration of compounds with antioxidant action, is particularly critical. We should give greater attention to the skin, considering it not as a waste product, but as an important part of the hazelnut due to its nutritional properties of primary relevance in the Mediterranean diet. We provide a detailed assessment of the nutritional properties of the hazelnut kernel, skin, and oil, evaluating nutrient compositions and possible modifications (increases or reductions) that occur during the roasting process or that depend on the production territory and origin.

## 1. Introduction

Dried fruits, such as hazelnuts, walnuts, almonds, pistachios, and pine nuts, are recommended as both breakfast and snack foods and are an integral part of the Mediterranean diet. This diet, traditionally widespread in Mediterranean countries such as Greece, Crete, Spain, and Italy, is characterized by a large consumption of cereals, vegetables, fruits, nuts (including hazelnuts), and extra virgin olive oil; the moderate consumption of fish and alcohol (mainly wine); and the reduced consumption of meat, its derivatives, and sweets [1]. 

Hazelnuts are part of the Mediterranean diet in various forms eaten whole (fresh or roasted), in the form of flour used to make bread or pasta, in sweets (e.g., nougat, chocolate bars, ice cream, and cakes), or as oil. Since the 1960s, the Mediterranean diet has undergone many changes due to the introduction of industrial products and cultural shifts caused by globalization. However, hazelnut consumption has continued in the form of industrial confectionery products and snacks of various kinds. Recent years have seen a resurgence in nut consumption due to increased awareness of their health benefits: for instance, consuming 30 g of nuts per day has been linked to reduced cardiovascular risk [2].

The positive effects of consuming nuts and other dried fruits in the Mediterranean diet can be identified in the particular composition of these fruits. In fact, even though they contain a high percentage of fats (60.8%) compared to most other foods, the content of saturated fatty acids is very low (4.5% in the kernel). Moreover, they are rich in monounsaturated fatty acids (MUFAs, making up 45.7% of the seed) and polyunsaturated fatty acids (PUFAs, constituting 7.9% of the seed). In addition, nuts are cholesterol-free, and their fat component contains measurable amounts of sterols, chemically related to cholesterol, part of a heterogeneous group of molecules called phytosterols [3]. These molecules have no nutritive role in plants, but they constitute their cell membranes, stabilizing the phospholipid bilayer. In the human gut, due to their high hydrophobicity, phytosterols interfere with cholesterol absorption, thus helping to lower its concentration in the blood [4]. 

Other beneficial effects of consuming nuts, particularly hazelnuts, may result from their high content of L-arginine, which, serving as a substrate for the synthesis of nitric oxide (the main regulator of vascular tone and consequently blood pressure), contributes to increased vascular reactivity and contributes to maintaining healthy blood pressure levels [5]. Additionally, their plant fiber content can contribute up to 5–10% of the daily requirement of the Mediterranean diet. 

Also of particular relevance to human health is the content of micronutrients such as folic acid (vitamin B9) and antioxidants (tocopherols and polyphenols) as well as their mineral salt composition. Rich in calcium, magnesium, and potassium, but low in sodium, they help lower blood pressure [2].

While all nuts share basic nutritional characteristics, each has nutritional peculiarities that vary based on whether they have skin and how they are prepared (raw or toasted). Hazelnuts can be eaten with or without the skin and either raw or toasted.

Turkey and Italy are the leading producers of hazelnuts (56% and 16%, respectively, according to 2017 FAO data) [6]. 

It is well known that Turkey grows a variety of hazelnuts, including Tabzon, Giresun, Ordu (which make up 40% of the production), Samsun, Akcakoca (20%), Tombul (30%), and other minor varieties. In Italy, on the other hand, several varieties are predominantly cultivated in Campania and Lazio, followed by Piedmont and Sicily. In Campania, Mortarella and San Giovanni varieties (30–35%) are grown for industrial consumption, whereas Tonda di Giffoni (TG) and other varieties are for fresh consumption. In Lazio, the Tonda Gentile Romana is mainly cultivated (85%); in Piedmont (Cuneo, Asti, and province), Tonda Gentile trilobata delle Langhe (TGL) is cultivated (90%), and finally, in Sicily, the Sicilian, Ghirara, Minnulara, and Lancinante varieties are cultivated. 

To date, numerous studies have been published on the nutritional composition of hazelnuts, but no review has comprehensively considered all the nutritional characteristics and how they vary based on factors such as roasting and the presence or absence of the skin.

The objectives of this narrative review will be as follows: An evaluation of the bromatological composition of hazelnuts and a comparison of the nutritional properties of raw versus roasted hazelnuts, seeking possible nutritional differences among varieties based on their production territories, such as Turkey, Italy, Chile, and New Zealand;An evaluation of the nutrients contained in hazelnut skin.An evaluation of the nutrients contained in hazelnut oil.

## 2. Materials and Methods

The present narrative review was performed following the steps by Egger et al. as follows [7]: (1)A working group was configured as follows: three operators skilled in clinical nutrition, with one acting as a methodological operator and two participating as clinical operators, and one operator, a molecular biologist, participating as a post-doc researcher for data collection and analysis.(2)The review question, based on considerations made in the abstract, was formulated as follows: “Evaluation of the bromatological composition of hazelnuts and comparison of the nutritional properties of raw versus roasted hazelnuts, with the aim of identifying possible nutritional differences among varieties from different production territories, such as Turkey, Italy, Chile, and New Zealand. Additionally, an evaluation of the nutrients contained in hazelnut skin and hazelnut oil.(3)Relevant studies were identified as follows: a research strategy was planned using PubMed, Cochrane Central Register of Controlled Trials (CENTRAL), and Web of Science as follows: (a) a definition of the keywords (nutritional composition, raw hazelnut, hazelnut skin, roasted hazelnut, hazelnut oil) was developed, followed by the definition of the interest field of the documents to be searched and grouped with quotation marks (“…”) used separately or in combination; (b) the Boolean AND operator, which allows for the establishment of logical relations among concepts, was used; (c) research modalities: advanced search was used; (d) limits: we included papers published in English in the last 25 years; (e) a manual search of reviews and individual articles on the nutritional composition of raw hazelnuts, hazelnut skin, and roasted hazelnuts in journals qualified in the Index Medicus was performed by senior researchers experienced in clinical nutrition.(4)The analysis was carried out in the form of a systematic review of the reports.

Based on the data from 45 scientific papers, including experimental papers and reviews, representative tables were created to analyze the bromatological composition of hazelnuts. These tables considered the origin and type of hazelnut processing and included data from a total of 27 articles in the study. Various nutrients, including essential and nonessential amino acids, proteins, carbohydrates, fiber, water, fatty and organic acids, phenols, flavonoids, mineral salts, vitamins, and antioxidant activity, were evaluated. The nutrients in raw hazelnuts (with skin), and roasted hazelnuts (both with and without skin) were analyzed. Furthermore, the nutritional properties of the hazelnut skin and the oil derived from hazelnuts were evaluated. Figure 1 shows the flow diagram.

## 3. Results

1. Evaluation of the bromatological composition of hazelnuts and comparison of the nutritional properties of raw versus roasted hazelnuts, aiming to identify potential nutritional differences among varieties depending on their production territories.

1.1. Essential and nonessential amino acids

This research was conducted using the keywords: “hazelnut compositional characteristics”, “nutritional composition of hazelnut”, “impact of roasting”, “influence of roasting”, “proteins in hazelnuts”, and “amino acids in hazelnuts”. A total of six articles were included.

Appendix A shows the nutritional characteristics of amino acids and proteins contained in (1) raw hazelnuts with skin (R_S) and (2) toasted hazelnuts without skin (T_WS) from Turkish hazelnuts grown in Turkey (Tombul).

1.2. Carbohydrates

This research was conducted based on the keywords: “hazelnut compositional characteristics” OR “nutritional composition of hazelnut” AND “impact of roasting” OR “influence of roasting” and “carbohydrates hazelnut” OR “sugars hazelnut” AND “fiber hazelnut”. One article was included.

Appendix A shows the nutritional characteristics of sugars, fiber, and carbohydrates contained in (1) raw hazelnuts with skin (R_S) and (2) toasted hazelnuts without skin (T_WS) from Turkish hazelnuts grown in Turkey (Tombul).

1.3a. Fatty acids in raw hazelnuts

This research was conducted based on the keywords: “hazelnut compositional characteristics” OR “nutritional composition of hazelnut” AND “impact of roasting” OR “influence of roasting” and “fats hazelnut”. A total of three articles were included.

Appendix A shows the nutritional characteristics of fats contained in (1) raw hazelnuts with skin (R_S) and (2) toasted hazelnuts without skin (T_WS) from Turkish hazelnuts grown in Turkey (Tombul).

1.3b. Fatty acid in roasted hazelnuts

This research was conducted based on the keywords: “hazelnut compositional characteristics” OR “nutritional composition of hazelnut” AND “impact of roasting” OR “influence of roasting” and “fats hazelnut” OR “fatty acid hazelnut”. A total of two articles were included.

Appendix A shows the fatty acid quantity contained in (1) raw hazelnuts with skin (R_S) and (2) toasted hazelnuts without skin (T_WS) from different origins: (2A) Turkish hazelnuts grown in Turkey (Tombul), (2B) Italian hazelnuts grown in Italy (TGT-TG), and (2C) Italian hazelnuts grown in Chile (TGT-Chile).

1.4. Organic acids, phenolic compounds in raw and roasted hazelnuts

This research was conducted based on the keywords: “hazelnut compositional characteristics” OR “nutritional composition of hazelnut” AND “impact of roasting” OR “influence of roasting” and “antioxidant activity hazelnut” OR “total phenols hazelnut”. A total of three articles were included. 

Appendix A shows the content of total phenols (free + bound), antioxidant activity, and individual phenols in (1) raw hazelnuts with skin (R_S) and (2) toasted hazelnuts without skin (T_WS) from different origins: (3A) Turkish hazelnuts grown in Turkey (Tombul), (3B) Italian hazelnuts grown in Italy (TGT-TG), and (3C) Italian hazelnuts grown in Chile (TGT-Chile).

1.5. Minerals and trace minerals in raw hazelnuts

This research was conducted based on the keywords: “hazelnut compositional characteristics” OR “nutritional composition of hazelnut” AND “impact of roasting” OR “influence of roasting” and “mineral salts hazelnut” OR “microelements hazelnut” AND “trace metal hazelnut”. A total of five articles were included. 

Appendix A shows the concentration analysis of mineral salts in Turkish Tombul raw hazelnuts with skin (R_S) and toasted hazelnuts without skin (T_WS).

1.6. Vitamins and folates in raw and roasted hazelnuts

This research was conducted based on the keywords: “hazelnut compositional characteristics” OR “nutritional composition of hazelnut” AND “impact of roasting” OR “influence of roasting” and “vitamins hazelnut” OR “carotenoids and tocopherol hazelnut”. A total of three articles were included. 

Appendix A shows the analyses of vitamins in Tombul and Viba Sweet Turkish hazelnuts by comparing raw hazelnuts with skin (R_S) to toasted hazelnuts without skin (T_WS) as well as a concentration analysis of mineral salts in Turkish Tombul raw hazelnuts with skin (R_S) to toasted hazelnuts without skin (T_WS).

1.7. Flavonoids in raw and roasted hazelnuts

This research was conducted based on the keywords: “hazelnut compositional characteristics” OR “nutritional composition of hazelnut” AND “impact of roasting” OR “influence of roasting” and “flavonoid hazelnut”. One article was included. 

Appendix A shows the analysis of flavonoid quantity contained in (1) raw hazelnuts with skin (R_S) and (2) toasted hazelnuts without skin (T_WS) from different origins: (A) Turkish hazelnuts grown in Turkey (Tombul), (B) Italian hazelnuts grown in Italy (TGT Piemonte-TG Campania), and (C) Italian hazelnuts grown in Chile (TGT-Chile).

2. Nutritional composition of hazelnut skin 

This research was conducted using the keywords: “hazelnut skin compositional characteristics”, “nutritional composition of hazelnut skin”, “roasted hazelnut skin”, “hazelnut husk”, “total polyphenols in hazelnut skin”, “phenolic acids in hazelnut skin”, “phenols/flavan-3-ols in hazelnut skin”, and “antioxidant activity of hazelnut skin”. A total of three articles were included. 

Appendix A presents the skin analysis of Giresun hazelnuts cultivated in Turkey, Tonda Gentile delle Langhe hazelnuts cultivated in Chile (TGL_Chile), Tonda Gentile delle Langhe hazelnuts cultivated in Italy (TG_Campania), and Tombul hazelnuts cultivated in Turkey. These findings have been added to Appendix A.

3. Hazelnut oil

This research was conducted using the keywords: “hazelnut oil”, “hazelnut oil compositional characteristics”, “nutritional composition of hazelnut oil”, “refined hazelnut oils”, “hazelnut oil lipid composition”, “oxidative stability of oils in hazelnuts”, and “antioxidative effects on refined hazelnut oil”. A total of six articles were included.

Appendix A presents the analysis of hazelnut oil from different origins: Turkish hazelnuts grown in Turkey (Tombul), and Italian Tonda Giffoni hazelnuts grown in Portugal (TG_Portugal) and in New Zealand (TG_New Zealand).

## 4. Discussion

1. This is an evaluation of the bromatological composition of hazelnuts, comparing the nutritional properties of raw versus roasted hazelnuts. The aim is to identify possible nutritional differences among the varieties depending on their production territories.

Before we begin by comparing hazelnuts with or without skins, it is good to explain what roasting involves and how nutritional values may vary based on the roasting process. Roasting is a crucial step in hazelnut processing for private use and for the confectionery industry. From a practical point of view, roasting has a positive effect on the safety of the food in terms of reducing aflatoxins and allergens. It increases the flavor and color of the fruit and leads to a desirable consistency in terms of crispness and friability. Currently, there are only a few studies that shed light on the effect of roasting on the bromatological composition of natural/raw hazelnuts and consequently, on its nutritional properties.

Thus, the effects of roasting can be assessed by evaluating the concentrations of micro and macronutrients, such as sugars, fatty acids, organic acids, and tocopherols [8]. Moreover, extending the time of roasting can result in increased antioxidant properties. However, these properties may vary depending on the roasting temperature, extraction protocols, and methods for assessing the concentration of antioxidants. Analyzing different articles, we find, in some cases, different opinions [9,10,11]. Moreover, there are few studies evaluating nutrients, such as phenolic profiles and antioxidant activity, in raw/natural, roasted, and roasted hazelnut skin that consider the territory of production. In Italy, the Tonda Gentile Trilobata hazelnut, formerly known as Tonda Gentile delle Langhe (TGL), is covered by the PGI (protected geographical indication, IGP) mark awarded by the European Union to products for which quality or another characteristic depends on geographical origin and whose processing takes place in a specified geographical area. The PGI Piedmont hazelnut is thus known as a high-quality hazelnut with good sensory characteristics and technological properties. Several parameters have been proposed to authenticate and trace these hazelnuts. Locatelli et al. proposed chemical and genotypic analysis to identify TGL produced in different geographical areas (Italy and Chile). Other parameters, such as RAPD markers, are able to distinguish different cultivars but are unable to distinguish whether the same cultivar was grown in different geographic areas. Since the main consumption of hazelnut is after roasting, Locatelli et al. wanted to look at TGL hazelnuts cultivated in Italy and Chile, TG hazelnuts from Italy, and Turkish Tombul hazelnuts, both raw and roasted (derived from commercial products such as pralines), with the aim of understanding whether there are significant variations in chemical parameters and whether these parameters allow identifying the cultivar and origin in terms of traceability of roasted hazelnuts. We also included Locatelli’s papers in our analysis [12].

1.1. Essential and nonessential amino acids

Hazelnuts are a source of essential and non-essential amino acids (a.a.). There are nine essential amino acids: histidine (His), isoleucine (Iso), leucine (Leu), lysine (Lys), methionine (Met), phenylalanine (Phe), threonine (Thr), tryptophan (Trp) and valine (Val). In their analysis of raw natural Tombul hazelnuts from Turkey, Alasalvar et al. found that the most concentrated essential amino acid is leucine (1.07 g/100 g), followed by valine, phenylalanine, and isoleucine [13]. Overall, total essential a.a. is about 4.68 g/100 g of hazelnut (Appendix A). Analysis of the essential amino acid content of Turkish roasted hazelnuts reveals that roasting results in a 10% decrease in the concentration of valine (from 40 to 33.3 g/100 g) while the total protein content does not change (Appendix A, Turkish hazelnut). 

Upon analyzing the eleven non-essential amino acids—alanine (Ala), arginine (Arg), asparagine (Asn), aspartic acid (Asp), cysteine (Cys), glutamic acid (Glu), glutamine (Gln), glycine (Gly), proline (Pro), serine (Ser), and tyrosine (Tyr)—it emerges that Glu (3.13 g/100 g) is the most abundant amino acid, followed by Arg (2.16 g/100 g) and Asp (1.52 g/100 g) [8]. Regarding asparagine (Asn) and glutamine (Gln), the authors did not analyze these amino acids. Overall, the total amount of non-essential a.a. is about 10.87 g/100 g of hazelnut (Appendix A).

Until now, there have been no articles describing the composition of all nine essential and all eleven non-essential amino acids together with the protein composition in roasted hazelnuts of Italian and Chilean origin.

We can conclude that the total amino acids in the Turkish Tombul hazelnut amount to 15.10 g/100 g (Appendix A).

It follows that the protein content in this hazelnut variety is 10.42 ± 8.7% [12,14] and its concentration is 16.39 ± 1.46 g/100 g [8,15]. The protein concentration in roasted hazelnuts is similar to that in raw hazelnuts, standing at 9.9 ± 8.6% (Appendix A). No descriptive articles were found on the amino acid composition of Italian hazelnuts, but it is believed that the concentrations of these a.a. are similar to Turkish hazelnuts.

1.2. Carbohydrates

The amount of carbohydrates, including simple and complex sugars as well as fiber, was analyzed in the Turkish hazelnut. Among simple sugars, glucose, fructose, myo-inositol, and sucrose were identified; whereas, among the complex sugars, raffinose and stachyose were evaluated. Raw hazelnuts show moderate levels of sucrose (2.67 g/100 g of product) compared to all other sugars present at lower concentrations (<0.5 g/100 g) (Appendix A). Among indigestible carbohydrates, hazelnuts contain a high concentration of fiber, which reaches an amount of 12.88 g/100 g in raw fruit (Turkish hazelnut). Hazelnuts are richer in insoluble fiber compared to soluble fiber (insoluble: 10.67 g/100 g and soluble: 2.21/100 g in Tombul) (Appendix A). It is believed that these values are comparable to those of the Italian hazelnut.

The total amount of carbohydrates observed is 17.30 g/100 g (Appendix A). However, no data is available regarding the carbohydrate concentration in roasted hazelnuts from different origins (Turkey, Italy, and Chile).

One gram of soluble fiber is known to decrease glucose levels and total cholesterol levels by 0.045 mmol/L and LDL cholesterol levels by 0.057 mmol/L in the blood. In addition, foods rich in soluble fiber reduce the risk of cancers, such as prostate cancer [16]. The insoluble fiber component of hazelnuts combats intestinal constipation. It has a sponge effect in the intestine where it absorbs water and increases the bulk of what is contained in the intestine and waste products, speeding their passage through the large intestine and expulsion, also reducing the risk of the onset of digestive diseases (cancers). Approximately 200 g of hazelnuts per day, added as a supplement to the diet, achieve the recommended daily fiber intake of 25–35 g/day [17] and have beneficial effects in patients with chronic degenerative diseases such as diabetes, hyperlipidemia, and obesity by controlling glucose, insulin, triglycerides, and cholesterol levels (absent in hazelnuts) by promoting satiety and interfering with carbohydrates and fats adsorption; moreover, this control is a benefit for preventing the onset of diabetes, cardiovascular diseases, and colon cancer [17,18].

1.3. Fatty acids in raw and roasted hazelnuts

1.3a. Fatty acids in raw hazelnuts

Considering that Turkish raw hazelnuts with skin contain about 62.8 ± 1.2 (%), 61.2 (g/100 g) of fats (Appendix A), consisting of only lipids and 0 g/100 g of cholesterol, and that very similar levels were observed in Italian hazelnuts (62.4%: TGT and TG hazelnuts) [12], the relative percentage of fatty acids was analyzed in raw hazelnuts of Turkish (Tombul) Italian (TGT Piemonte, Campania) or Chilean origin by evaluating the 2015 scientific papers by Locatelli et al. and Schlörmann. The data were collected in Appendix A.

Starting with raw hazelnuts (raw with skin), it was observed that Turkish hazelnuts contain about 63.6 ± 1.7 (%), 61.2 (g/100 g) of fats (Appendix A). 

In these papers, the authors first extracted lipids from the hazelnuts and then transesterified them under alkaline conditions by adding NAPCH3 and methanol as reagents; this procedure led to producing fatty acid methyl esters (FAME) separated by a silica capillary column. Semi quantification of individual fatty acids was calculated as the relative percentage of the total area of all FAME using GC Solution Software (from Methods et al., 2015) [14]. If necessary, we calculated the average relative %, SD, and CV from the reported data sets. 

A total of twelve different fatty acids, including saturated and mono/poly-unsaturated acids, were extracted and quantified from the Turkish hazelnut (Appendix A); we also reported the quantification of five of them: *palmitic acid* (C16:0), *stearic acid* (C18:0), the saturated fatty acid (SFA) *oleic acid* (C18:1Δ9), the monounsaturated fatty acid (MUFA) *vaccenic acid* (C18:1Δ11), and, finally, the poly-unsaturated fatty acid (PUFA) *linoleic acid* (C18:2 *n*-6). These acids were analyzed by authors both in the Italian (Appendix A) and Chilean (Appendix A) varieties.

We observed that Tombul Turkish raw hazelnuts contain about 5.33 ± 0.32% of *palmitic acid* (C16:0), about 2.4 ± 0.6% of *stearic acid* (C18:0), 81.7 ± 1.3% of *oleic acid* (C18:1Δ9), and 9.2 ± 0.2% of *linoleic acid* (C18:2 *n*-6) (Appendix A).

Regarding the two major Italian varieties analyzed as raw hazelnuts, we observed that both TGT_ Piemonte and Tonda di Giffoni (TG_Campania), have an average percentage of 5.9 ± 0.4 of *palmitic acid* (C16:0), 3 ± 0.2% of *stearic acid* (C-18:0), 81.4 ± 1.7% of *oleic acid* (C18:1Δ9), and 8.9 ± 2.9% of *linoleic acid* (C18:2 *n*-6)(Appendix A) [12]. 

In both Turkish and Italian raw hazelnuts grown in Turkey and Italy, respectively, the average relative percentage of oleic (C18:1 Δ9), linoleic (C 18:2), and palmitic (C16:0) fatty acids was higher compared to the other acids, as indicated in Appendix A (Average column) [12,14].

The same chemometric analysis on the fatty acids profile performed on Italian hazelnut samples grown in Chile (TGT_Chile) shows values for the three fatty acids at 6.9 ± 0.0% (C16:0), 80.8 ± 0.4% (C18:1 Δ9), and 7.7 ± 0.3% (C18:2) (Appendix A), which were similar to those obtained from the hazelnuts cultured in Turkey or Italy described in Appendix A. 

By comparing all the raw hazelnuts of different origins, we observed a slight increase in *linoleic acid* (C18:2) content in Tombul Turkish raw hazelnuts compared to the Italian ones (Italy or Chile growth), and a slight increase in *palmitic acid* (C16:0) in the Italian TGT hazelnuts grown in Chile compared to both the Turkish and the Italian hazelnuts grown in Italy; however, these reductions were not significant. Schlormann et al. in their 2015 study reported the total saturated fatty acids (SFAs), monounsaturated fatty acids (MUFAs), and polyunsaturated fatty acids (PUFAs) in all the hazelnut varieties analyzed. Considering only five fatty acids analyzed in all Turkish, Italian, and TGT_Chilean hazelnut varieties, the total SFAs were 7.73%/8.85%/9.9%, respectively. The total MUFAs were 83.04%/82.93%/82.4% and the total PUFAs were 9.15%/8.88%/7.73%%. (Appendix A). 

Finally, Schlörmann et al. (2015) aimed to distinguish the nutritional benefits of hazelnuts, which are rich in fatty acids [14]. They compared the relative percentage of fatty acids in hazelnuts with other nuts, such as almonds, macadamia nuts, pistachios, and walnuts. We observed that, among monounsaturated fatty acids, hazelnuts contain the most oleic acid (C 18:1 Δ9): Hazelnut (82.6%) > Almonds (66.8%) > Macadamia nuts (58.4%), Pistachios (53.8%) > Walnuts (12.3%). This result could have importance for human health, considering the benefits of MUFAs [14]. When comparing raw hazelnuts with peeled, roasted hazelnuts, it can be seen that the total saturated fatty acids (SFAs) are slightly lower in raw hazelnuts, corresponding to 7.8% in raw and 8.2% in roasted hazelnuts. Meanwhile, monounsaturated fats account for 83.3% in raw and 82.4% in roasted hazelnuts, and polyunsaturated fats account for 9.3% in raw and 9.4% in roasted hazelnuts (Appendix A).

1.3b. Fatty acid in roasted hazelnuts

Published data show that roasted hazelnuts have a similar percentage value of fats (64.1%) compared to raw hazelnuts (62.8%) because the weight is more concentrated and there is no fat in the skin [12]. According to reported data sets regarding skinless roasted hazelnuts, we analyzed the same five different types of unsaturated fatty acids (C16, C18:0, C18:1Δ9, C18:1Δ11, C18:2) in the raw fruit looking for their relative % after roasting. The Turkish hazelnut variety, Tombul, contains about 81.2 ± 1.2% of oleic fatty acid (C18:1Δ9), about 9.4 ± 1.3% of C18:2 fatty acid, and 5.6 ± 0.3% of palmitic acid (C16:0). The others are present in lower percentages [12,14]. No significant variations emerge regarding the composition of these fats (Appendix A). The Italian hazelnut in the two varieties analyzed (TGT or delle Langhe_Piemonte, Tonda di Giffoni TG_Campania), compared to Turkish hazelnuts, has an average percentage of C18:1 Δ9 fatty acid equal to 81.5 ± 1.7 and C18:2 fatty acid equal to 8.6 ± 2.1 (Appendix A) [12]. The same analysis performed on samples of Italian hazelnut (TGT_Piemonte) but grown in Chile confirms the values obtained for the two fatty acids analyzed and reveals that there are no significant differences regarding fatty acid composition in the samples analyzed (Appendix A). Therefore, it is concluded that there are no significant differences in fatty acid composition in Turkish hazelnut versus Italian hazelnut, whether the latter is grown in the country of origin or in other countries such as Chile. Data reported in the literature do not show a significant change in the percentage of fat in hazelnuts after roasting, compared to raw hazelnut (Appendix A) [12].

The roasting process does not significantly interfere with FAME [12,14], and, thus, it does not affect the fatty acid profile. Moreover, no significant differences were observed among the different roasted hazelnut varieties in terms of total SFAs, MUFAs, and PUFAs analyzed (Appendix A).

Regarding individual fatty acids, oleic acid is one of the omega-9 fatty acids (ω-9) that contributes, with polyunsaturated ones, to the regulation of human blood cholesterol levels by promoting an increase of HDL, reducing the risk of cardiovascular diseases; to the regulation of hormone levels; and to the protection of neuronal cells from senescence. After roasting/toasting and removing the skin (Toasted Without Skin_T WS) of Tombul, TGT_Piedmont, TG Campania, and TGT Chile hazelnuts, oleic acid was found to have the highest concentration, compared to the others, as observed in the raw kernels described before (relative % average, Appendix A). 

The fact that oleic acid as well as linoleic and palmitic acids were not reduced in the skinless roasted hazelnuts, compared to the raw ones with skin, nor in all four analyzed hazelnut varieties, demonstrates that both raw and roasted hazelnuts are important for the Mediterranean diet. In conclusion, there are no significant differences in fatty acid composition in raw and toasted hazelnuts from Turkey or Italy, regardless of whether the latter is grown in the country of origin or in other countries such as Chile (Appendix A). 

1.4. Organic acids, phenolic compounds in raw and roasted hazelnuts

1.4a. Organic acids, phenolic compounds in raw hazelnuts

Organic acids, also known as carboxylic acids, have a COOH group. They are capable of forming hydrogen bonds and very frequently participate in reactions in which they lose their OH group to form esters, anhydrides, or amides. Organic acids include gallic, protocatechuic, caffeic, ferulic, coumaric, salicylic, syringic, vanillic, 4-hydroxybenzoic, and sinapic acids. In the raw Turkish hazelnuts with a peel that were analyzed, protocatechuic acid (5.31 μg/g) was found to be the most concentrated among these acids, followed by gallic (2.39 μg/g) and caffeic (1.99 μg/g) acids (Appendix A) [12]. 

These acids, together with ferulic, coumaric, and sinapic acids, have also been analyzed in other scientific studies [9,19]. In a 2015 study, Locatelli et al. extracted phenols with methanol from fat-deprived hazelnut powder and tested them using the Folin Ciocalteu (1965) assay. In Shahidi et al., phenolic acids were obtained from ethanol and methanol extracts, which were analyzed with the Shimadzu HPLC system and the Folin Ciocalteu assay (1965); the concentration was measured to quantify the phenolic acids in μg/g of extract. The most concentrated acid in the raw fruit is *p*-coumaric acid (208 µg/g). Gallic acid, on the other hand, is most abundant in the hull, the hazelnut shell, and the green cover of the hazelnut (Appendix A) [19]. In Pelvan’s 2018 study, the extraction was performed as described in Robbins’ 2015 study using ethyl ether quantified by the Folin Ciocalteu assay and expressed per µg/100 g hazelnut in the study and transformed by us to µg/g in order to have uniform and comparable measurements. In this study, gallic acid is more concentrated (6.79 µg/g hazelnut) than the others analyzed. In the 2007 study by Shahidi et al., gallic acid is 127 µg/g, which is 18 times higher than in Pelvan’s study. The dominance of different phenolic acids across various studies could be attributed to the extraction/solvent methods used as well as environmental factors such as harvest time, cultivation and drying methods, season, and storage and handling conditions (Appendix A) [9].

Italian raw hazelnuts with skin (TGT, TG) have a higher concentration of protocatechuic acid (5.54 ± 0.33 μg/g), compared to gallic acid (2.18 ± 0.46 μg/g) and caffeic acid (1.88 ± 0.35 μg/g), as observed in Turkish hazelnuts [12]. We also observed an approximate 25% increase in gallic and caffeic acid concentrations in TGT Piemonte compared to TG Campania (Appendix A).

Considering the growing territory of Italian raw hazelnuts, in the 2015 study by Locatelli et al., the TGT hazelnuts grown in Chile show a significant increase in the concentration of protocatechuic acid (6.59 µg/g) compared to the other acids (Appendix A), although not significant when compared to the same acid evaluated in the fruit grown in Italy (5.54 ± 0.33 µg/g) or Turkey (5.31 µg/g). (Appendix A)

In conclusion, organic acids, such as protocatechuic, gallic, and caffeic acids, are similarly concentrated between Italian hazelnuts (TGT or TGT in the Chile variety) and Turkish hazelnuts; moreover, these acids are highly concentrated, giving raw hazelnuts a relevant importance in the Mediterranean diet due to their antioxidant, antimicrobial and anticancer organic nutrients. Gallic acid is also known for its anti-obesity properties, while caffeic acid is used in the clinical management and treatment of cancer and neurological diseases [20].

1.4b. Organic acids, phenolic compounds in roasted hazelnuts

In the 2018 study by Pelvan et al. on hazelnuts of Turkish origin (Tombul), the raw hazelnut with the peel had approximately 1.45 times higher phenolic acid content than the roasted hazelnut without the peel. This shows that by removing the peel after roasting, a good portion of some organic acids is lost [9]. In the roasted fruit without husk, there was a significant decrease in gallic acid (0.63 µg/g) and sinapic acid (1.37 µg/g), and an increase, although not significant, in the concentration of the acids syringic and vanillic compared to the raw hazelnut with husk (Appendix A) [9].

In the 2015 study by Locatelli et.al, the roasting temperature was considered. The Turkish hazelnut with husk was analyzed after roasting at different temperatures, and it was observed that, in general, roasting at temperatures above 110 °C significantly increased the concentration of gallic acid (2.39 µg/g versus 12.51 µg/g at 160 °C and 12.36 µg/g at 180 °C); however, no differences in terms of the final concentration of this acid were found between the two temperatures, 160 and 180 °C, to which the hazelnut was subjected.

In the same study, the organic acid compositions (gallic, protocatechuic, and caffeic) were analyzed by comparing the two Italian hazelnut varieties (TGT_Piedmont and TG_Campania) with the Italian hazelnut grown in Chile (TGT_Chile) after roasting (Appendix A). It can be observed that, regardless of the roasting temperature, gallic acid in Italian hazelnuts grown in Italy increased approximately 4-fold in the roasted hazelnut without skin (8.17 ± 3.1 µg/g) compared to the raw hazelnut (2.18 ± 0.46 µg/g). Additionally noteworthy is the significant difference between the two Italian cultivars (TGT_Piedmont, TG_Campania) grown in Italy (Appendix A). 

Performing the same analysis on the TGT_Chile hazelnut, it is shown that gallic acid increases from 2.98 µg/g to 9.70 µg/g in the peel-less roasted hazelnut compared to the peel-roasted hazelnut. The high roasting temperature leads to a greater increase in the concentration of gallic acid: at 180° C, from 2.98 µg/g to 10.83 µg/g. Protocatechuic and caffeic acids decrease, but not significantly. These changes, especially for the TGT_Chile hazelnut, need further experimental confirmation by increasing the number of biological replicates. (Appendix A).

1.5a. Minerals and trace minerals in raw hazelnuts

Hazelnuts are rich in minerals, including calcium (Ca), magnesium (Mg), potassium (K), and trace minerals including phosphorus (P). The most concentrated mineral in Turkish hazelnut (Tombul) is K (675.5 mg ± 81.1/100 g), followed by Ca (171 mg/100 g), Mg (161 mg/100 g), and trace minerals such as P (332 mg ± 29.4/100 g) and Na (2.93 mg/100 g). (Appendix A). Among oligominerals, there are trace amounts of selenium (0.03 mg/100 g). (Appendix A). Selenoproteins protect cell membranes from oxidative stress. Medical studies show that selenium may prevent the occurrence of cancers (breast, colon, lung, and prostate) Finally, oligominerals such as Al, As, Cd, Pb, and Ag are present in trace amounts (≤10 mg/100 g) (Appendix A) [8,14,15,21]. 

Studies show that the concentration of these trace elements in hazelnuts may vary depending on variety, geographic origin, climate, fertilizer, and soil composition. 

A diet that relies on 100 g of hazelnuts per day provides a good percentage of the recommended intake of required metals per day in the adult male: about 38% of Fe for raw hazelnuts and about 28% for roasted hazelnuts, about 67% of Mg for raw hazelnuts and about 56% for roasted hazelnuts, and about 17% of Ca for raw hazelnuts and about 14% for roasted hazelnuts [22].

1.5b. Minerals and trace minerals after roasting

The composition of minerals in Turkish Tombul hazelnuts after roasting at different temperatures has also been analyzed in several scientific papers [12,14,23,24,25]. The values obtained in the following articles were averaged by keeping the concentration as mg/100 g of hazelnut. Roasting results in a 20% decrease in the concentration of calcium (141.9 mg/100 g) and magnesium (135.8 mg/100 g), and a 90% decrease in potassium (6.04 mg/100 g). It also results in a 90% increase in sodium (661 mg/100 g) (Appendix A). A 90% increase in silicon (9.83 ± 12.57 mg/100 g) is observed as well as a 50% increase in chromium (0.02 ± 0.02 mg/100 g). A reduction in aluminum, selenium, cobalt, manganese, and cadmium is also observed (Appendix A). 

However, these variations are not homogenous between the elements analyzed, and this may depend on the technique used and the fact that sampling and analysis were carried out in different laboratories. 

Currently, there are no articles describing the concentration of these minerals in Italian roasted hazelnuts.

Analyzing the data on the minerals contained in hazelnuts, we can conclude that the main macro-nutrients (Ca, P, Mg, K) decrease in roasted hazelnuts without peel, apart from sodium, which increases.

As far as the toxic trace elements (Al, As, Cd, Pb, Ag) are concerned, they decrease in the peel-less roasted hazelnut. 

1.6. Vitamins and folates in raw and roasted hazelnuts

1.6a. Vitamins and folates in raw hazelnuts

The vitamin content of hazelnuts in terms of vitamins B1, B2, B5, B6, C, E, biotin, niacin, folate, and beta-carotene was analyzed. Turkish hazelnut is rich in vitamins E active form (α-tocopherols) (27.9 ± 11.3 mg/100 g) and C (5.54 ± 0.2 mg/100 g) followed by niacin and B5 (≤2 mg/100 g) and the others B vitamins (≤1 mg/100 g) (Appendix A). Among the 11 types of nuts (hazelnut, walnut, Brazilian walnut, almond, pistachio, etc.) hazelnut contains the highest concentration of vitamin E, folate, and biotin. In addition, 40 g of hazelnuts per day provides 100% of the daily requirement of vitamin E [8,14,25]. Vitamin E (α-tocopherol) is a lipid-soluble phenolic antioxidant. Through the phenolic component, which donates an H atom to free radicals, it has an antioxidant action, preventing cancers, atherosclerosis, and diabetes. About 100 g of hazelnuts per day provides 60% of the folate (200 µg total daily folate) recommended for adults. 

No analysis of the vitamin content of the Italian hazelnut has been revealed, but it is thought to be similar to that of the Turkish hazelnut. 

1.6b. Vitamins and folates after roasting 

Following roasting, there is no significant decrease or increase in the analyzed vitamins (vit: B1, B2, B6, B12, folate) (Appendix A) [12,14,21,23,24,25]. Vitamin E (α-tocopherols), on the other hand, reports a non-significant decrease of 20% in peel-less roasted hazelnuts.

1.7. Flavonoids in raw and roasted hazelnuts

1.7a Flavonoids in raw hazelnuts

The term antioxidants (the most important being flavonoids) refers not only to a chemical term but also to a context of cellular oxidative stress. Any molecule can be an oxidant or a reductant and is determined by the reduction potential of the molecule with which it reacts. An antioxidant is defined as a molecule that limits oxidative stress by reacting ‘non-enzymatically’ with a reactive oxidant molecule. An antioxidant enzyme is a protein that limits oxidative stress by catalyzing a redox reaction with a reactive oxidant. At the endogenous level, our body has enzymatic and non-enzymatic components that aim to prevent oxidative stress by preventing the formation of free radicals or by eliminating free radicals that cause oxidative stress. These include glutathione and thyrotoxin. Furthermore, a diet rich in antioxidants is believed to prevent the onset of oxidative stress. Among the various tests, the FRAP test measures water-soluble and fat-soluble antioxidants and allows one to assess the body’s antioxidant intake. 

Raw Turkish hazelnuts contain significant levels of flavonoids, such as quercetin (7.50 µg/g), catechin (7.32 µg/g), and myricetin (7.05 µg/g) [12,25] (Appendix A). Raw Italian hazelnuts (TGT_Piemonte, TG_Campania) cultivated in Italy had slightly lower, but similar, antioxidant values as Turkish hazelnuts, including quercetin (6.39 ± 1.35 µg/g), catechin (6.84 ± 2.35 µg/g) and myricetin (6.24 ± 1.68 µg/g) (Appendix A). 

TGT-Piemonte hazelnuts cultivated in Chile had 10–15% higher levels of all analyzed flavonoids (catechin, epicatechin, quercetin, myricetin, and kaempferol) than TGT hazelnuts cultivated in Italy (Appendix A). However, TG-Campania hazelnuts also had approximately 20% more catechins (9.55 µg/g) than all other hazelnuts analyzed (Italian and Turkish) (Appendix A).

1.7b. Flavonoids in roasted hazelnuts

The unroasted Turkish hazelnut shows a 30% increase in catechins compared to the raw Turkish hazelnut, increasing from 7.32 µg/g to 10.57 µg/g. For the other flavonoids, there is a slight non-significant reduction. (Appendix A).

Roasted Italian hazelnuts without skin double the catechin content compared to raw Italian hazelnuts from 6.84 µg/g to 13.86 µg/g. Quercetin increases in roasted hazelnuts by about 15% while the other flavonoids show similar values. (Appendix A).

Catechin in TGT-Piemonte hazelnuts cultivated in Chile is similar to that in the hazelnuts cultivated in Italy (double the value in the roasted hazelnuts compared to the raw ones). All other flavonoids decrease non-significantly in roasted, skinless hazelnuts. (Appendix A).

2. Nutritional composition of hazelnut skin 

As already mentioned, the hazelnut fruit consists of a 2–3 mm thick (hard) shell inside which is a fruit (stone) with a hull (husk). The fruit can be eaten raw with the husk (peel) or roasted (without peel). The peel covering the fruit accounts for about 2.5% of the fruit weight and, very often, in the confectionery industry, it is discarded after roasting [26]. Several studies have examined the peel with the aim of understanding whether certain micronutrients could be preserved after roasting the peel. Recent studies introduced the concept that hazelnut hulls may have beneficial properties for the body due to the presence of tocopherols, amino acids, and polyphenols, which may have positive effects on cardiovascular health, e.g., by helping to protect LDL-cholesterol from oxidation [2]. 

Del Rio et al. analyzed the peel after roasting four varieties of hazelnuts of Turkish origin (Akakoca, Giresun, Ordu, and Trabzon), three of Italian origin (Tonda di Giffoni and Mortarella_Campania, Tonda Gentile delle Langhe_ Piemonte, and Tonda Gentile Romana_Lazio) and one Italian variety cultivated in Chile (Tonda Gentile delle Langhe coltivata¬_Cile) [27]. These products were supplied by the company Soremartec Italia (Alba). 

Ordu and Akakoca are the varieties that contain the most flavon-3-ols, while prodelphinidins reach the highest level in the skin of TGL. Catechins are contained in 80% of the flavon-3-ols. The Ordu variety contains the highest catechin content, while TGL contains the highest level of epicatechin and its gall form. The Ordu variety also contains the highest concentration of phenolic acids, while the lowest is found in the Akakoca variety.

The skin of the Italian hazelnut cultivated in Chile shows the lowest concentration of all flavanols. Using the Folin Ciocalteu assay, the Akakoca variety has the highest polyphenol content. The peel contains the highest antioxidant content compared to other foods, such as chocolate and wine. A diet rich in antioxidants, including flavonoids, reduces inflammation and increases vascular elasticity. 

In the 2018 study by Pelvan et al., the Turkish Tombul hazelnut was examined, identifying the content of phenolic acids in raw hazelnuts with skin, in the hazelnuts after roasting without skin, and individually in the hazelnut skin. The following phenolic acids (gallic, protocatechuic, salicylic, ferulic, syringic, vanillic, 4-hydroxybenzoic, caffeic, o-coumaric, and sinaptic) were analyzed under the three different conditions. Roasted hazelnut skin has a higher content of total phenolic acids, about 489 compared to natural hazelnuts, and 710 times higher than roasted hazelnuts without skin. Raw hazelnuts contain 1.45 times more phenolic acids than its counterpart after roasting without the peel. Most of the phenolic components are contained in the skin of the roasted hazelnut. Gallic acid is the predominant compound in the husk. It constitutes 98.5% of the total phenolic acid content in roasted hazelnut husks [9].

The studies by Shaididi et al. and by Alsalvar C. in 2007 show that gallic acid is the most abundant phenolic acid present in the hazelnut husk, while del Rio’s 2011 study reports that protocatechuic acid is the most abundant in the husk. The difference may be due to the extraction method, environmental factors, storage, and handling. Furthermore, in the same study, the antioxidant activity was analyzed using different quantification methods (DPPH scavenger activity): the amount of extract required to reduce the initial DPPH radical by 50%, expressed in mg per sample, was evaluated. Roasted hazelnuts with peel show a higher antioxidant activity than peel-less or raw roasted hazelnuts. These data demonstrate the high content of phenolic compounds. 

In Appendix A, we wanted to compare the two Italian hazelnuts, Tonda Gentile delle Langhe grown in Italy (TGL_Piemonte) and Tonda di Giffoni grown in Italy (TG_Campania), with Tonda Gentile delle Langhe hazelnuts grown in Chile (TGL_ Chile) and the two Turkish hazelnuts—Tombul hazelnuts grown in Turkey, which we chose as a reference for this study, and Giresun hazelnuts grown in Turkey. We chose to add the Giresun to compare with another Turkish hazelnut because, in the 2011 study by Del Rio et al., the Tombul quality was not analyzed. Therefore, we chose to include Giresun hazelnuts, which, being analyzed in the same study as the Italian hazelnuts with the same method, offer interesting data that is more comparable, compared to the other two studies by Shahidi et al. in 2007 and Pelvan et al. in 2018, which did not analyze the Italian hazelnuts.

Appendix A shows the phenols/flavan-3-ols. In this table, one can see a significant difference in all the values for the Tonda Gentile delle Langhe grown in Chile, which presents significantly lower values than the same hazelnuts grown in Italy, but also compared to the other hazelnuts grown in Italy, Giresun turca and Tonda di Giffoni. 

The hazelnut with the highest values in all phenols/flavan-3-ols, except for procyanidin trimers, grow in Italy.

The effects of raw Turkish hazelnuts and their bioproducts, such as the hull (after roasting), the shell, the leaf cover enclosing the hazelnut (leaf cover), and the plant leaf (tree leaf), on scavenging hydrogen peroxide to prevent lipid oxidative stress, LDL-cholesterol oxidation, and DNA oxidation was assessed [19]. Compared to the whole fruit and catechins, hazelnut bioproducts, such as the hull and leaf shell of the hazelnut, have been found to show a higher hydrogen-peroxide-scavenging capacity, proving useful in combating oxidative stress.

The hazelnut skin and green fruit shell are able to inhibit LDL-cholesterol oxidation at a concentration of 50 ppm (99 and 93%, respectively) compared to the fruit, shell, and leaf of the hazelnut. However, all hazelnut extracts were much more effective in inhibiting such oxidation than the catechins used as a reference [19]. Furthermore, at the DNA level, the hull shows significantly greater inhibition of DNA oxidation than the other extracts and the fruit. Furthermore, all hazelnut bioproducts have a greater antioxidant effect than the fruit. Finally, it can be concluded that hazelnuts and their bioproducts, including the fruit, exhibit antioxidant and protective activity due to their ability to scavenge hydroxyl radicals up to a concentration of 50 ppm.

The hull, currently considered a by-product of the hazelnut, could be used in food with health benefits (nutraceuticals) due to its high concentration of phenolic and antioxidant compounds.

Therefore, different strategies for husk recovery in the roasting stages could be considered for introduction into hazelnut processing.

3. Hazelnut oil

In addition to hazelnut kernels and flour, oil can also be extracted from the hazelnut seed. 

Hazelnut oil has not been extracted for edible purposes until recently. In the past, hazelnuts were generally consumed as an appetizer [28].

Classification of hazelnuts: 

Shape: Hazelnut cultivars can be grouped into three main groups: round, conical, and almond shape. 

Quality and yield: The Giresun type is accepted as a high-quality hazelnut. The hazelnuts have a high oil yield, easy processing, and good flavor. The other type of hazelnuts, the Levant type, has a lower oil yield than the Giresun type [29].

In Turkey, there are several companies that process mainly low-quality hazelnuts with the aim of producing edible oil in this way [30].

How it is obtained

The oil is obtained by cold mechanical pressing of the fruit and separates the oily component from the creamier one. This process maintains its organoleptic properties. 

Hazelnut oil is extracted by chemical and physical techniques from unshelled hazelnuts. Both raw and roasted hazelnuts can be used in oil processing. Products with different aromas and colors can be obtained from roasted hazelnuts.

The roasting conditions may vary depending on the manufacturer, but the roasting temperature and time are generally between 100–160 °C for 10–60 min.

Depending on the variety of hazelnut, the study by Özkan G. at al in 2016 found that as the temperature and roasting time increased, the oil yield changed. 

The increase in roasting time and temperature is a relevant parameter for oil yield [29]. 

Another relevant parameter of hazelnut oil is the flavor, which can be significantly increased by roasting the hazelnuts before pressing; in fact, hazelnut oils sold on the market can be raw products from roasted or unroasted hazelnuts, refined oils from unroasted hazelnuts, or mixtures of these. The roasting of hazelnuts prior to pressing has a minimal effect on the composition of the oil [31].

Oil characteristics

Hazelnuts are rich in mono- and polyunsaturated fatty acids, particularly oleic and linoleic acids. The oxidative stability of hazelnut oils is the strongest among vegeTable Soils, with a high content of oleic acid and phenolic substances [29]. 

What it contains and what properties it possesses

The oil is rich in vegeTable Sfats including omega-3. Hazelnut oil, like the fruit from which it is derived, contains many vitamins, including group A vitamins, group B vitamins, and vitamin E. It is a good source of selenium, which helps prevent the aging of cells, and calcium, which helps strengthen the skeleton, hair, and nails. The oil contains many lipids and flavonoids, belonging to the group of polyphenols with anti-inflammatory action, which can help the body prevent viral infections and the onset of certain types of cancer (Appendix A).

Hazelnut oil contains phytosterols that reduce the risk of cardiovascular disease. Due to its high omega-3 content, the oil supports the reduction of blood cholesterol. Overall, hazelnut oil is very nutritious, promotes tissue regeneration, and stimulates circulation, and for these reasons, its use in the Mediterranean diet is recommended [13].

A diet high in MUFAs (monounsaturated fatty acids) tends to increase HDL cholesterol and reduce triacylglycerol (TAG) concentrations. Therefore, hazelnuts, as an excellent source of MUFAs, may prove beneficial. In addition to MUFAs, it contains polyunsaturated fatty acids (PUFAs) and tocotrienol. These components have been reported to reduce plasma concentrations of total and LDL cholesterol. Furthermore, a diet rich in MUFAs instead of carbohydrates can favorably influence cardiovascular disease (CVD) risk and has positive effects on atherosclerosis. In addition, phytosterols (α -sitosterol) reduce the risk of certain cancers and CVD and improve immune function. Hazelnut oil, which is an excellent source of tocopherols, has been shown to reduce the risk of coronary heart disease (CHD). In recent years, the health effects of vitamin E isoforms (tocopherols and tocotrienols) have been well documented. TAGs are a group of non-polar lipids that account for almost 100% of the total non-polar lipids in hazelnut oil. TAGs are increasingly used in the food industry as a tool to assess the quality and authenticity of vegeTable Soils, in particular the adulteration of olive oil with hazelnut oil. In summary, hazelnut oil is a good source of natural and bioactive antioxidants, reflecting its nutraceutical potential in various food and specialty applications. Furthermore, hazelnut oil, a rich source of MUFAs (oleic acid), phytosterols (α -sitosterol), and vitamin E, can be considered as a supplement for daily diet planning to reduce the risk of CHD by lowering total, LDL, VLDL, and TAG cholesterol levels and increasing HDL cholesterol levels [13].

How it is used

In cooking, it is used in high-end catering, given its high cost, and is used to flavor mixed salads and fruit salads. 

In confectionery, it is used for cakes, tarts, and biscuits as a substitute for butter or margarine. 

In cosmetics, it is used as a nourishing agent for epithelia in creams or as a moisturizer for the whole body, particularly indicated in cases of skin irritation, on sensitive and dry skin, in cases of eczema and erythema, and even in children; this is due to its composition, which has anti-inflammatory properties and is able to confer flexibility and elasticity to the skin, even after only a few applications of topical use. 

It can, therefore, be concluded that hazelnut oil is particularly suitable for both nutritional and cosmetic use, given its important beneficial properties for the health of the body [8]. 

Counterfeiting

The chemical composition of hazelnut oils depends on the geographical origin, the variety of hazelnut, and the extraction process. The lipid profile of hazelnut oil appears to be very similar to that of olive oil. It is not uncommon that hazelnut oil is used to stretch olive oil. The article by Benitez-Sànchez P.L. et al. in 2003 compared different oils and identified certain triacylglycerides that allow hazelnut oils to be distinguished from olive oils [32].

Processing

Refining is applied to crude oils to remove excessive amounts of undesirable compounds such as free fatty acids, waxes, polar lipids, oxidation products, metal ions, and pigments.

This is conducted through a series of steps: degumming, neutralization, bleaching, deodorization, and winterization. Not all these steps necessarily have to be applied to all crude oils, sometimes the degumming and winterization steps can be bypassed depending on the composition and condition of the oil. In some cases, neutralization can be performed together with the deodorization step.

During oil refining, in addition to undesirable compounds such as free fatty acids and oxidation products, some desirable compounds such as antioxidants and triglycerides are also removed to some extent. Although this seems an ineviTable Sprocess for most seed oils, the disadvantages of the refining process must be discussed. Not only does the refining process result in the loss of valuable phytochemicals, but it also results in the formation of potentially harmful substances such as chloropropanols and trans fatty acids [30].

Processing issues

Oils are important carriers of some fat-soluble bioactive compounds. Refining causes a significant decrease in the bioactivity of hazelnut oil. This reduction is measured as a decrease in tocopherols, phenolic compounds, and the loss of carotenoids (lutein and zeaxanthin). Consequently, the antioxidant capacity was reduced. In contrast, the change in oxidative stability in the neutralization phase led to a marked increase in oxidative stability, measured by the Rancimat method, compared to crude oil. However, the deodorizing step caused a slight decrease in oxidative stability, probably as a result of the partial removal of tocopherols at this stage. Considering the possible other drawbacks of edible oil refining, such as the formation of chloropropanols, glycidyl esters, trans fatty acids, and other health-threatening compounds, the need for new approaches in this area is evident [30].

The aim of refining is to remove unwanted impurities [e.g., free fatty acids (FFA), phospholipids, and color pigments] while damaging beneficial constituents (e.g., tocopherols and sterols) as little as possible and minimizing oil loss.

Chemical refining can cause a 10–20% reduction in the tocopherol content of vegeTable Soils, presumably due to the absorption of soaps formed during alkaline treatments. Tocopherols perform the important function of protecting the oil from oxidation. Sterols are minor components of all-natural fats and oils and comprise the remainder, which is essentially hydrocarbons. During neutralization, a considerable part of the phytosterols is transferred by liquid-liquid partition to the soap mass. In addition, bleaching and high-temperature steam refining or deodorization remove some of the sterols [28].

During processing, especially in heat treatment, flavor constituents are often degraded or evaporated, causing a deterioration in product quality that may not be accepted by consumers. In the 2022 study by Kesen, S. et al., it was observed that the refining process reduced the amounts of aroma constituents by about eight times [33].

Storage 

The storage of hazelnut oil is rather difficult because it is rich in unsaturated fatty acids. They are susceptible to oxidation and deteriorate easily with the reaction of oxygen with air, especially during long storage periods. Lipid peroxidation is the most common deterioration factor in vegeTable Soils. Oxidation reacts with lipids and a series of free radical chain reactions leads to complex chemical changes. Oxidation leads to an unpleasant, rancid taste, loss of nutrition, and the formation of compounds that are potentially toxic to human health [34]. 

There are a number of methods to prevent oxidation of oils. The most common and effective method is the addition of antioxidant agents in food formulations. In general, some synthetic antioxidant agents such as butylated hydroxytoluene (BHT), butylated hydroxyanisole (BHA), tertiary butyl hydroquinone (TBHQ), and propyl gallate (PG) are used to prevent oxidation in the food industry. However, some scientific research has highlighted the possibility of toxic effects on human health. For this reason, natural antioxidant compounds that are safe and effective and alternatives to synthetic antioxidants are being sought; in recent years, there has been an increasing trend towards the use of natural antioxidants in food formulations. In general, various herbs and spices show antioxidant properties because they have antioxidant compounds in their structure. Therefore, they provide significant and effective protection in preventing or delaying lipid peroxidation. Their antioxidant characteristics are related to certain phenolic compounds. These compounds possess redox properties and are able to act as reducing agents or donors of hydrogen atoms and as free radical quenchers. Phenolic compounds are synthesized in substantial quantities (0.5–1.5%) and widely distributed in plants. These compounds have been found to possess antioxidant and free radical scavenging activity in food [34].

Hazelnut oil vs. olive oil

Hazelnuts are a rich source of oleic acid (approximately 80%), which has been associated with beneficial health effects, and compared to olive oil, it has the advantage of having less saturated fatty acids. Hazelnuts also contain several phytosterols, generally in higher amounts than most olive oil samples, which appear to be important bioactive compounds as they can inhibit the intestinal absorption of cholesterol. Although some differences between cultivars have been noted, further data are needed to confirm whether the composition of the lipid fraction of hazelnuts differs enough to say whether some cultivars are better for health [35].

Comparison of oils from different cultivars

In Appendix A, we compared hazelnut oil from Tombul Turkish hazelnuts grown in Turkey and Italian Tonda Giffoni hazelnuts grown in Portugal and in New Zealand (TG Campania).

Linoleic acid (18:2 ω6) in the Turkish hazelnut was 8.94 ± 0.13% vs. a higher concentration of 11.41 ± 1.58% in the TG Campania. Linoleic acid (18:3 ω3), on the other hand, has an overlapping value for the two cultivars of 0.11%. Beta-Sitosterol in Tombul has a lower concentration than in the TG Campania equivalent: 123.51 ± 15.62 mg/100 g vs. 154.10 ± 12.01 mg/100 g. The relevant presence of total tocopherol should be emphasized. In Tombul, it is 39.78 ± 5.20 mg/100 g, and in TG Campania, it is 44.72 mg/100 g.

As pointed out above, the saturated fatty acids in the hazelnut are 7.11 ± 1.23 g/100 g in the Turkish hazelnut and 7.81 g/100 g in the Italian quality. The total MUFAs in the Turkish hazelnut is 60.70 ± 31.88 g/100 g, and in the Italian cultivar, it is 81.75 g/100 g. The total PUFAs in the Tumbul is 8.98 ± 0.01 g/100 g lower than in the Campania TG at 11.55 ± 1.63 g/100 g.

## 5. Conclusions

This review is based on the narrative analysis of data contained in 27 peer-reviewed scientific articles published from 1997 to 2022, with the aim of evaluating possible differences in the bromatological composition of hazelnuts depending on their different processing stages, derivatives, and origin. Specifically, the different processing stages considered in this study were (i) raw hazelnuts with skin, (ii) roasted hazelnuts, (iii) skin obtained after roasting, and (iv) hazelnut oil.

Hazelnuts, like other oilseeds, have assumed a very important role in nutrition from ancient times to the present day. In contemporary society, it is mainly consumed as an ingredient in complex food products; in fact, it is used whole, in the form of grains or flour in a great many confectionery products, and in some non-confectionery products. In addition, hazelnut oil can be obtained from it for use in cooking as a condiment. The reason for their success in our diet should be sought in their high energy content (672 kcal), mainly due to their high lipid content (55–69.9 g/100 g). In contrast to animal fats, those contained in hazelnut are mainly monounsaturated. Hazelnut contains low levels of saturated fatty acids and cholesterol. Because of these characteristics, hazelnut is considered beneficial for human health and able to prevent cardiovascular disease. Hazelnuts contain protein (12.7–17.5 g/100 g), carbohydrates (15.9 g/100 g), and fiber (1.8–6.2 g/100 g). Soluble fiber is able to decrease cholesterol, while insoluble fiber promotes intestinal transit of waste products derived from food, fighting constipation. The concentration of amino acids is not sufficient for our daily needs, and it is necessary to supplement the diet with other foods. In addition, this fruit contains organic acids, including gallic acid and protocatechuic acid. Regarding the composition related to minerals and trace minerals, hazelnut contains high levels of potassium (K), followed by calcium (Ca), magnesium (Mg), phosphorus (P), and selenium (Se) in trace amounts. Hazelnut also contains high levels of antioxidants, including α-tocopherols, quercetin, catechin, myricetin, and antioxidant-acting vitamins such as vitamins E and C. Due to the presence of these molecules, the consumption of hazelnuts or their derivatives contributes to the reduction of free radical formation and limits oxidative stress to the body’s cells. 

Roasting has a positive effect on food safety in terms of reducing aflatoxins and allergens, increasing the flavor and color of the fruit, and leading to a desirable texture in terms of crunchiness and crispness. In this process, hazelnuts are cooked at temperatures between 100 °C and 180 °C, and there is a loss of most of the water they contain. It has been observed that this process results in a decrease in the concentration of metals such as K, Ca, and semi-metals such as P; while there is an increase in sodium (Na). Roasting does not result in a significant change in amino acids, vitamins, folates, or fatty acids, but there is a significant increase in gallic acid. However, few studies shed light on the effect of roasting on the bromatological composition of hazelnuts and, consequently, the effects of roasting on their nutritional properties. It is believed that this topic can be further explored by performing studies aimed at evaluating the effects of roasting temperature variations on the bromatological composition of hazelnuts.

In the studies concerning the skin or the husk, it was examined after the hazelnut was roasted. When analyzing the bromatological composition of the husk, which accounts for 2.5% of the total weight of the hazelnut, it is observed that it has the highest amount of antioxidants compared to other foods such as chocolate and wine. From the data published by Shahidi et al. in 2007, it was found that, compared with the whole fruit and shell, the husk was better at fighting oxidative stress [19]. However, the husk is considered a by-product and waste product of hazelnut processing. Peeling off spontaneously after roasting, the husk is removed after this processing step and is usually not included in the final products. Based on its nutritional properties, mainly regarding its antioxidant compounds, it is clear that the husk can be retained in all stages of hazelnut processing or alternatively consumed independently or as a basic ingredient for more complex foods. Based on the data analyzed, we think that further studies should be conducted evaluating nutrients such as amino acids and minerals in the husk. 

Oil, as a product derived from hazelnuts, is gradually finding popularity because of its excellent nutritional properties. In fact, it reflects the bromatological composition of the whole fruit in terms of fat-soluble components, such as vitamin E, high levels of monounsaturated fats, and antioxidant compounds. It is also rich in omega 3.

The oil can be obtained from raw and roasted hazelnuts and extracted by mechanical pressure or solvents. The data on the oil reveal that it was obtained from hazelnuts cultivated in Turkey, New Zealand, Spain, and Chile; however, no data emerges on the bromatological composition of hazelnut oil from hazelnuts grown on Italian soil. Furthermore, it is noted that, in some cases, the authors do not clarify the type of processing the hazelnuts underwent. For this reason, further studies should be carried out to assess the bromatological composition of the oil by broadly analyzing the oil derived from hazelnuts cultivated in Italy and other countries, focusing on organic acids, fatty acids, and antioxidants, the latter being highly present in hazelnuts.

The demonstration that the hull has a higher content of antioxidant compounds than other foods known for these properties elevates this component of the hazelnut, still considered waste, to a foodstuff with very important nutritional properties. Further studies are needed to explore the effects of roasting on the nutritional properties of hazelnuts, as well as to assess the composition of hazelnut oil from different sources.

## Figures and Tables

**Figure 1 nutrients-15-02421-f001:**
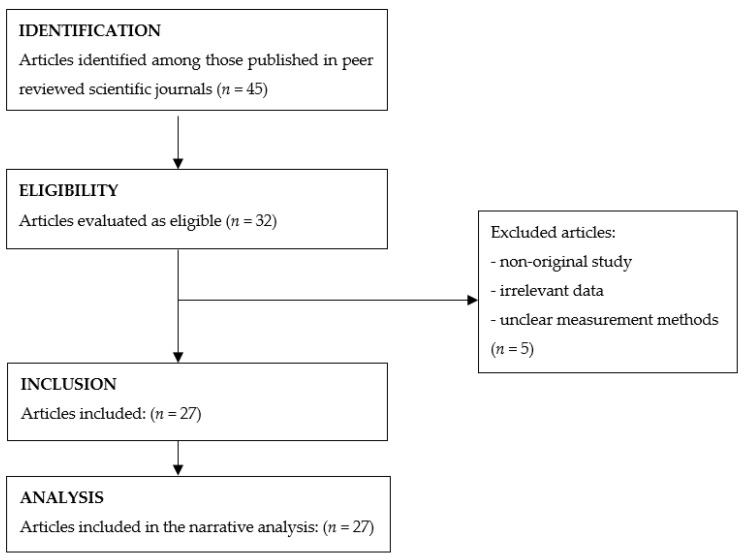
Flow diagram.

## Data Availability

Data is contained within the article or Appendix A.

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
