# Peer review of "Phytoextracts for Human Health from Raw and Roasted Hazelnuts and from Hazelnut Skin and Oil: A Narrative Review"

_nutrients, 2023, doi:10.3390/nu15112421_

Round 1
Reviewer 1 Report
It was a nice attempt to examine the nutritional composition of hazelnut and how may that vary with regions and food processing method. I have two major comments for authors to revise their manuscript.
First, the review paper must be re-structured according to PRISMA checklist. Looking down the list, there are many parts being merged or content not being explained well enough. For example, has the review being retrospectively registered? When was the literature search being conducted? What are the specific eligibility criteria to include or exclude papers? What are the details that have been extracted from each included study? From Figure 1, it’s unclear why the screening starts from full-text articles instead of abstracts, which results in so few potentially eligible papers for screening? And the questions can go on and on, which needs to be resolved by restructuring the paper, and add the PRISMA checklist in supplementary information so readers can understand where the corresponding content can be found.
Second, results and discussion are merged which make the key information not easy to find. It’ll be much better to separate the content with reference to PRISMA, presenting the study characteristics first, then going to the main body of results, then moving on to the discussion. In the discussion part, please make it clear how each research question raised in the introduction has been addressed. More importantly, authors have to be clear on whether the varied nutritional composition in hazelnut is substantial enough to influence the dietary recommendation. For example, if the nutritional value is significantly reduced under certain conditions, that should be avoided so that the general population can still adhere to the Mediterranean Diet.
Nil
Author Response
Dear Reviewer,
we have revised the text according to your suggestions.
Best regards,
the authors
It was a nice attempt to examine the nutritional composition of hazelnut and how may that vary with regions and food processing method. I have two major comments for authors to revise their manuscript.
First, the review paper must be re-structured according to PRISMA checklist. Looking down the list, there are many parts being merged or content not being explained well enough. For example, has the review being retrospectively registered? When was the literature search being conducted? What are the specific eligibility criteria to include or exclude papers? What are the details that have been extracted from each included study? From Figure 1, it’s unclear why the screening starts from full-text articles instead of abstracts, which results in so few potentially eligible papers for screening? And the questions can go on and on, which needs to be resolved by restructuring the paper, and add the PRISMA checklist in supplementary information so readers can understand where the corresponding content can be found.
ANSWER: Dear reviewer, thank you for the suggestion. Certainly using the PRISMA standards would be interesting, but, as reported in the title, in the abstract and in the materials and methods paragraph, this is a narrative and non-systematic review and for this reason we have not followed the PRISMA checklist, but we have followed the indications for the narrative review, as specified in the materials and methods. We have chosen to carry out a narrative review, certainly less impactful than the systematic review, but this choice is specifically due to the topic we wanted to deal with, which made it impossible to manage the topic as a systematic review. Not being a systematic, but narrative review, we have no application process on PROSPERO (International prospective register of systematic reviews).
Second, results and discussion are merged which make the key information not easy to find. It’ll be much better to separate the content with reference to PRISMA, presenting the study characteristics first, then going to the main body of results, then moving on to the discussion. In the discussion part, please make it clear how each research question raised in the introduction has been addressed. More importantly, authors have to be clear on whether the varied nutritional composition in hazelnut is substantial enough to influence the dietary recommendation. For example, if the nutritional value is significantly reduced under certain conditions, that should be avoided so that the general population can still adhere to the Mediterranean Diet.
ANSWER we have divided results and discussion, given that the narrative review allows this choice. Regarding the specific request on how much the nutritional composition can influence the dietary recommendation, we have added some specific sentences on this topic.
Reviewer 2 Report
The author of this narrative review aims to evaluate the bromatological composition of hazelnuts, compare the nutritional properties of raw vs. roasted hazelnuts, and assess the nutrients contained in hazelnut skin and oil. The review includes 27 scientific articles that measured and reported the concentrations of macro- and micro-nutrients in hazelnuts from different processing methods, geographical areas, and varieties around the world. The study concludes that the composition of hazelnuts is influenced by their variety and cultivation territory, and that different processing steps can significantly affect the concentration of specific nutrients. Additionally, the review highlights the critical importance of hazelnut skin, which contains a high concentration of antioxidant compounds and should be considered an important part of the nut's nutritional value. Overall, the topic of this review is valuable, as hazelnuts are an important food source with nutritional benefits that are still being explored. However, the manuscript would benefit from further editing to improve clarity and flow. For example, some sentences are wordy and could be shortened without losing meaning.
There are several comments to help the authors improve their manuscript.
the review could be expanded to include more research on the nutritional properties of hazelnuts and their potential health benefits.
the conclusion also notes the need for further studies to explore the effects of roasting on the nutritional properties of hazelnuts, as well as to assess the composition of hazelnut oil from different sources.
The writing and formatting of the paper are not up to standard, and the authors should optimize the paper according to the general requirements of a review article.
The images included in this work are visually unappealing and may benefit from optimization.
The language in this manuscript requires SIGNIFICANT improvement. For example, several grammatical errors were present in the abstract section, indicating a lack of attention to detail. The authors should thoroughly check for and correct grammatical errors throughout the paper.
"looking at" should be "looking for" or "examining".
"such as Turkey, Italy, and Chile" needs a comma before "and".
"in hazelnuts" should be "in the hazelnuts".
"around the world were measured and reported" should be "around the world, and their concentrations were measured and reported."
"its bromatological composition" should be "their bromatological composition" since hazelnuts are plural.
"it was found" should be "we found".
"particularly critical is the removal of the skin" should be "The removal of the skin is particularly critical".
"nutritional properties of primary relevance in the Mediterranean diet" should be "nutritional properties that are of primary relevance in the Mediterranean diet".
"by evaluating nutrient compositions" should be "by evaluating the nutrient compositions".
"increase or reduction" should be "increases or reductions".
"roasting process" should be "roasting process," since it's the end of the sentence and the start of a new idea.
Author Response
Dear Reviewer,
we have revised the text according to your suggestions.
Best regards,
the authors
The author of this narrative review aims to evaluate the bromatological composition of hazelnuts, compare the nutritional properties of raw vs. roasted hazelnuts, and assess the nutrients contained in hazelnut skin and oil. The review includes 27 scientific articles that measured and reported the concentrations of macro- and micro-nutrients in hazelnuts from different processing methods, geographical areas, and varieties around the world. The study concludes that the composition of hazelnuts is influenced by their variety and cultivation territory, and that different processing steps can significantly affect the concentration of specific nutrients. Additionally, the review highlights the critical importance of hazelnut skin, which contains a high concentration of antioxidant compounds and should be considered an important part of the nut's nutritional value. Overall, the topic of this review is valuable, as hazelnuts are an important food source with nutritional benefits that are still being explored. However, the manuscript would benefit from further editing to improve clarity and flow. For example, some sentences are wordy and could be shortened without losing meaning.
There are several comments to help the authors improve their manuscript.
the review could be expanded to include more research on the nutritional properties of hazelnuts and their potential health benefits.
ANSWER: some sentences have been added on this topic in the text
the conclusion also notes the need for further studies to explore the effects of roasting on the nutritional properties of hazelnuts, as well as to assess the composition of hazelnut oil from different sources.
ANSWER: The sentence has been added
The writing and formatting of the paper are not up to standard, and the authors should optimize the paper according to the general requirements of a review article.
ANSWER: see the answers to reviewer 1.
The images included in this work are visually unappealing and may benefit from optimization.
ANSWER: images have been enhanced
Comments on the Quality of English Language
The language in this manuscript requires SIGNIFICANT improvement. For example, several grammatical errors were present in the abstract section, indicating a lack of attention to detail. The authors should thoroughly check for and correct grammatical errors throughout the paper.
ANSWER: An English native speaker, an expert in scientific revisions, reread the text and made the necessary changes; the document certifying the revision is attached.
"looking at" should be "looking for" or "examining". DONE
"such as Turkey, Italy, and Chile" needs a comma before "and". DONE
"in hazelnuts" should be "in the hazelnuts". DONE
"around the world were measured and reported" should be "around the world, and their concentrations were measured and reported." DONE
"its bromatological composition" should be "their bromatological composition" since hazelnuts are plural. DONE
"it was found" should be "we found". DONE
"particularly critical is the removal of the skin" should be "The removal of the skin is particularly critical". DONE
"nutritional properties of primary relevance in the Mediterranean diet" should be "nutritional properties that are of primary relevance in the Mediterranean diet". DONE
"by evaluating nutrient compositions" should be "by evaluating the nutrient compositions". DONE
"increase or reduction" should be "increases or reductions". DONE
"roasting process" should be "roasting process," since it's the end of the sentence and the start of a new idea. DONE
Round 2
Reviewer 1 Report
Since authors intended not to publish a systematic review, I have on further comments on the current content.
Nil
Author Response
Since authors intended not to publish a systematic review, I have no further comments on the current content.
Answer: thanks for your previous suggestions, which improve the quality of the paper.
Reviewer 2 Report
The revised version is an improvement, but there are still a few areas that could use some refinement for grammar and clarity:
For example, in the abstract section,
"The objectives of this narrative review will be as follows:" - It's usually best to write in the present tense when talking about the content of the paper, so "The objectives of this narrative review are as follows:" would be better.
"evaluation of hazelnut bromatological composition" - "Bromatological" is a term that refers to the study of food in terms of its nutritional value and it doesn't need to be hyphenated with a number. This should be "evaluation of the bromatological composition of hazelnuts".
"raw v/s roasted hazelnut looking for possible nutritional differences among the varieties depending on their production territory, 20 such as Turkey, Italy, and Chile;" - The abbreviation "v/s" isn't standard English. Use "vs." or "versus" instead. The sentence could be more clear if rephrased. Something like: "comparison of the nutritional properties of raw versus roasted hazelnuts, taking into account potential differences among varieties from different production territories such as Turkey, Italy, and Chile;".
"This review included 27 scientific articles in which the concentrations of macro- and micro-nutrients in hazelnuts subjected to different processing or from different geographical areas or belonging to different varieties around the world, and their concentrations were measured and reported." - The structure of this sentence is a bit confusing. Consider rephrasing to something like: "This review incorporates 27 scientific articles that measured and reported the concentrations of macro- and micro-nutrients in hazelnuts. These hazelnuts were subjected to different processing methods, originated from various geographical areas, or belonged to different varieties."
"Our results showed that the different varieties and the territory where the hazelnuts is cultivated influence their bromatological composition, and we found that different processing steps can largely influence the concentration of specific nutrients." - The phrase "the hazelnuts is cultivated" should be "the hazelnuts are cultivated" for correct subject-verb agreement.
"The removal of the skin is particularly critical, which contains a very high concentration of compounds with antioxidant action." - This sentence would be clearer if rephrased to: "The removal of the skin, which contains a very high concentration of compounds with antioxidant action, is particularly critical."
"Greater attention should be given to the skin by considering it not a waste product, but as an important part of the hazelnut with nutritional properties that are of primary relevance in the Mediterranean diet." - Consider rephrasing for clarity: "We should give greater attention to the skin, considering it not as a waste product, but as an important part of the hazelnut due to its nutritional properties of primary relevance in the Mediterranean diet."
"A detailed assessment of the nutritional properties of the hazelnut kernel, skin and oil by evaluating the nutrient compositions and possible modifications (increases or reductions) during roasting process, or depending on production territory and origin, has been provided." - This sentence is somewhat confusing. Consider rephrasing to: "We provide a detailed assessment of the nutritional properties of the hazelnut kernel, skin, and oil, evaluating nutrient compositions and possible modifications (increases or reductions) that occur during the roasting process or that depend on the production territory and origin."
In the introduction section,
"fruits, nuts and hazelnuts" - Hazelnuts are a type of nut, so including "nuts" and "hazelnuts" in the same list could be a bit redundant. You could rephrase this to "fruits, nuts (including hazelnuts)," to make it clear that hazelnuts are part of the broader category of nuts.
"as flours used to make bread or pasta, as sweets (e.g., nougat, chocolate bars, ice cream, and cakes), or as oil." - It would be clearer to say "in the form of flour used to make bread or pasta, in sweets (e.g., nougat, chocolate bars, ice cream, and cakes), or as oil."
"In recent decades (since the 1960s) the Mediterranean diet has undergone many changes, with the entry of industrial products and cultural change due to globalization." - This sentence is fine, but it might be clearer to say "Since the 1960s, the Mediterranean diet has undergone many changes due to the introduction of industrial products and cultural shifts caused by globalization."
"Recent years have seen a further resurgence of nut consumption due to awareness of the beneficial effects of their consumption on health: an intake of 30 g per day of nuts reduces the cardiovascular risk [2]." - This is correct, but you might consider rephrasing for clarity: "Recent years have seen a resurgence in nut consumption due to increased awareness of their health benefits: for instance, consuming 30 g of nuts per day has been linked to reduced cardiovascular risk [2]."
"In fact, although composed of a very high percentage of fats (60.8 %) compared to most other foods, the content of saturated fatty acids is very low (4.5% in the kernel)." - Rephrase for clarity: "In fact, even though they contain a high percentage of fats (60.8%) compared to most other foods, the content of saturated fatty acids is very low (4.5% in the kernel)."
"Moreover, monounsaturated fatty acids (MUFA, 45.7% of the seed) and polyunsaturated fatty acids (PUFA, 7.9 % of the seed) are very abundant." - This is fine, but could be rephrased for clarity: "Moreover, they are rich in monounsaturated fatty acids (MUFAs, making up 45.7% of the seed) and polyunsaturated fatty acids (PUFAs, constituting 7.9% of the seed)."
"In the human gut, phytosterols, due to their high hydrophobicity, interfere with the absorption of cholesterol, helping to lower its concentration in the blood [4]." - Rephrase for clarity: "In the human gut, due to their high hydrophobicity, phytosterols interfere with cholesterol absorption, thus helping to lower its concentration in the blood [4]."
"Other beneficial effects of consuming nuts, particularly hazelnuts, may result from their high content of L-arginine, which, serving as a substrate for the synthesis of nitric oxide (the main regulator of vascular tone and consequently blood pressure), contributes" - This sentence appears to be incomplete. You could finish it like this: "... contributes to maintaining healthy blood pressure levels." Or, if you have specific information on how L-arginine contributes to health, you could include that instead.
"Of particular relevance to human health is also the content of micronutrients such as folic acid (vitamin B9) and antioxidants (tocopherols and polyphenols) and their mineral salt composition." - The phrase "is also the content" is awkward. A smoother phrasing might be: "Also of particular relevance to human health is the content of micronutrients such as folic acid (vitamin B9) and antioxidants (tocopherols and polyphenols), as well as their mineral salt composition."
"Even if all nuts have shared basic nutritional characteristics, however, each has nutritional peculiarities that also vary based on whether or not they have the skin and how they are prepared (raw or toasted)." - The sentence could be smoother without "even if" and "however": "While all nuts share basic nutritional characteristics, each has nutritional peculiarities that vary based on whether they have skin and how they are prepared (raw or toasted)."
"The hazelnut has the characteristic of being able to be eaten with and without skin and raw or toasted." - This could be simplified to: "Hazelnuts can be eaten with or without skin, and either raw or toasted."
"It is well known that the hazelnut varieties grown in Turkey are the following with different percentage spread: Tabzon, Giresun, Ordu (40%), Samsun, Akcakoca (20%), Tombul (30%) and other minor varieties." - This sentence could be made clearer: "It is well known that Turkey grows a variety of hazelnuts, including Tabzon, Giresun, Ordu (which make up 40% of the production), Samsun, Akcakoca (20%), Tombul (30%), and other minor varieties."
"In Italy, on the other hand, we find several varieties that are cultivated the most, in Campania and Lazio followed by Piedmont and then Sicily." - This sentence could be improved: "In Italy, on the other hand, several varieties are predominantly cultivated in Campania and Lazio, followed by Piedmont and Sicily."
"To date, various research has been published on the nutritional composition of hazelnuts, but there is no review that considers all the nutritional characteristics and the variations of the same related to the roasting and the presence or absence of the skin." - This could be rephrased for clarity: "To date, numerous studies have been published on the nutritional composition of hazelnuts, but no review has comprehensively considered all the nutritional characteristics and how they vary based on factors such as roasting and the presence or absence of skin."
"Evaluation of Hazelnut bromatological composition and comparison of nutritional properties of raw v/s roasted hazelnut looking for possible nutritional differences among the varieties depending on their production territory, such as Turkey, Italy, and Chile." - Consider rephrasing for clarity: "Evaluation of the bromatological composition of hazelnuts and comparison of the nutritional properties of raw versus roasted hazelnuts, seeking possible nutritional differences among varieties based on their production territory, such as Turkey, Italy, and Chile."
Remember to maintain the same tense throughout your objectives. As these are objectives for a future review, using the future tense might be more appropriate, e.g., "The objectives of this narrative review will be to evaluate..." etc. This would depend on the overall context of your work.
In the Materials and Methods section,
"A working group was configured as follows: three operators skilled in clinical nutrition, of whom one acting as a methodological operator, two participating as clinical operators; one operator as Molecular Biologist participating as post-doc Researcher for data collection/analysis." - Consider revising for clarity: "A working group was configured as follows: three operators skilled in clinical nutrition, with one acting as a methodological operator, two participating as clinical operators, and one operator, a Molecular Biologist, participating as a post-doc researcher for data collection and analysis."
"The revision question on the basis of considerations made in the abstract was formulated as follows: “Evaluation of Hazelnut bromatological composition and comparison of nutritional properties of raw v/s roasted hazelnut looking for possible nutritional differences among the varieties depending on their production territory, such as Turkey, Italy, and Chile; Evaluation of nutrients contained in hazelnut skin; Evaluation of nutrients contained in the hazelnut oil.”" - Again, consider revising for clarity: "The review question, based on considerations made in the abstract, was formulated as follows: “Evaluation of the bromatological composition of hazelnuts and comparison of the nutritional properties of raw versus roasted hazelnuts, with the aim of identifying possible nutritional differences among varieties from different production territories, such as Turkey, Italy, and Chile. Additionally, an evaluation of the nutrients contained in hazelnut skin and hazelnut oil."
"Starting from the data published in 45 scientific papers, including experimental papers and reviews, it was possible to create representative tables of the analysis of the bromatological composition of hazelnuts based on the origin and type of hazelnut processing by including a total of 27 articles in the study." - A slight rephrase for clarity: "Based on the data from 45 scientific papers, including experimental papers and reviews, representative tables were created to analyze the bromatological composition of hazelnuts. These tables considered the origin and type of hazelnut processing, and included data from a total of 27 articles in the study."
"Different nutrients, including essential and nonessential amino acids, proteins, carbohydrates, fiber, water, fatty and organic acids, phenols, flavonoids, mineral salts, vitamins, antioxidant activity, were evaluated." - Consider revising for clarity: "Various nutrients, including essential and nonessential amino acids, proteins, carbohydrates, fiber, water, fatty and organic acids, phenols, flavonoids, mineral salts, vitamins, and antioxidant activity, were evaluated."
"Nutrients in the raw hazelnut with skin, roasted without skin/with skin were analyzed, furthermore, the nutritional properties of the skin and finally of the oil derived from the hazelnut were evaluated." - Consider revising for clarity: "The nutrients in raw hazelnuts (with skin), roasted hazelnuts (both with and without skin) were analyzed. Furthermore, the nutritional properties of the hazelnut skin and of the oil derived from hazelnuts were evaluated.”
In the results section,
"Evaluation of Hazelnut bromatological composition and comparison of nutritional properties of raw v/s roasted hazelnut looking for possible nutritional differences among the varieties depending on their production territory "Suggestion: "Evaluation of hazelnut bromatological composition and comparison of the nutritional properties of raw versus roasted hazelnuts, aiming to identify potential nutritional differences among varieties depending on their production territories."
"from Turkish grown in Turkey (Tombul)." Suggestion: "from Turkish hazelnuts grown in Turkey (Tombul)."
"This research was conducted based on the keywords: “hazelnut compositional characteristics” OR “nutritional composition of hazelnut” AND “Impact of roasting” OR “Influence of roasting” and “proteins hazelnut” OR “aminoacids hazelnut”." Suggestion: "This research was conducted using the keywords: “hazelnut compositional characteristics”, “nutritional composition of hazelnut”, “Impact of roasting”, “Influence of roasting”, “proteins in hazelnuts”, and “amino acids in hazelnuts”."
"from different origin" Suggestion: "from different origins"
"concentration analysis of mineral salts in the Turkish Tombul raw with skin (R_S) hazelnut and toasted without skin (T_WS) hazelnut."Suggestion: "concentration analysis of mineral salts in the Turkish Tombul raw hazelnut with skin (R_S) and toasted hazelnut without skin (T_WS)."
"This research was conducted based on the keywords: “skin hazelnut compositional characteristics” OR “nutritional composition of skin hazelnut” AND “skin roasted hazelnut” OR “husk hazelnut” AND “total polyphenol skin hazelnut” AND “phenolic acids skin hazelnut” AND “phenols / flavan-3-ols skin hazelnut” AND “skin hazelnut antioxidant activity”." Suggestion: "This research was conducted using the keywords: “hazelnut skin compositional characteristics”, “nutritional composition of hazelnut skin”, “roasted hazelnut skin”, “hazelnut husk”, “total polyphenols in hazelnut skin”, “phenolic acids in hazelnut skin”, “phenols/flavan-3-ols in hazelnut skin”, and “antioxidant activity of hazelnut skin”."
"TABLE 7 (A;B;C;D:E) shows the skin analysis of Giresun hazelnut cultivated in Turkey, Tonda Gentile delle Langhe in Chile (TGL_Chile), Tonda Gentile delle Langhe in Italy (TG_Campania) and Tombul in Turkey was added to Table B. " Suggestion: "TABLE 7 (A;B;C;D:E) presents the skin analysis of Giresun hazelnuts cultivated in Turkey, Tonda Gentile delle Langhe in Chile (TGL_Chile), Tonda Gentile delle Langhe in Italy (TG_Campania), and Tombul in Turkey. These findings have been added to Table B."
"This research was conducted based on the keywords: “oil hazelnut” OR “oil hazelnut compositional characteristics” OR “nutritional composition of oil hazelnut” AND “refined hazelnut oils” AND “oil hazelnut Lipid composition” AND “oxidative stability of oils in hazelnuts” OR “Antioxidative effects on refined hazelnut oil”."Suggestion: "This research was conducted using the keywords: “hazelnut oil”, “hazelnut oil compositional characteristics”, “nutritional composition of hazelnut oil”, “refined hazelnut oils”, “hazelnut oil lipid composition”, “oxidative stability of oils in hazelnuts”, and “antioxidative effects on refined hazelnut oil”."
"TABLE 8 shows the analysis of hazelnut oil from different origin: Turkish grown in Turkey (Tombul), Italian hazelnut Tonda Giffoni grown in Portugal (TG_Portugal) and in New Zealand (TG_New Zealand)." Suggestion: "TABLE 8 presents the analysis of hazelnut oil from different origins: Turkish hazelnuts grown in Turkey (Tombul), and Italian Tonda Giffoni hazelnuts grown in Portugal (TG_Portugal) and in New Zealand (TG_New Zealand)."
In the discussion section,
"Evaluation of Hazelnut bromatological composition and comparison of nutritional properties of raw v/s roasted hazelnut looking for possible nutritional differences among the varieties depending on their production territory "Suggestion: "This is an evaluation of the bromatological composition of hazelnuts, comparing the nutritional properties of raw versus roasted hazelnuts. The aim is to identify possible nutritional differences among the varieties depending on their production territories."
"Few studies currently shed light on the effect of roasting on the bromatological composition of the natural /raw hazelnut and consequently on its nutritional properties."Suggestion: "Currently, there are only a few studies that shed light on the effect of roasting on the bromatological composition of natural/raw hazelnuts and consequently, on its nutritional properties."
"Thus, it is possible to assess the effects of roasting by evaluating the concentration of micro and macro nutrients such as sugars, fatty acids, organic acids and tocopherols [8]."Suggestion: "Thus, the effects of roasting can be assessed by evaluating the concentrations of micro and macronutrients such as sugars, fatty acids, organic acids, and tocopherols [8]."
"We included also Locatelli’s papers for our analysis”,Suggestion: "We also included Locatelli’s papers in our analysis. "
"Analyzing raw natural hazelnut of the Tombul variety (Turkey) Alasalvar et al. show that the most concentrated essential a.a. is Leucine (1.07g/100g), followed by Valine, Phenylalanine and Isoleucine."Suggestion: "In their analysis of raw natural hazelnuts of the Tombul variety from Turkey, Alasalvar et al. found that the most concentrated essential amino acid is Leucine (1.07g/100g), followed by Valine, Phenylalanine, and Isoleucine."
"Analyzing the eleven non-essential amino acids: Alanine (Ala), Arginine (Arg), Asparagine (Asn), Aspartic acid (Asp), Cysteine (Cys), Glutamic acid (Glu), Glutammin (Gln), Glycine (Gly), Proline (Pro), Serine (Ser), Tyrosine (Try) it emerges that Glu (3.13 308 g/100 g) is the most abundant a.a, followed by Arg (2.16 g/100 g) and Asp (1.52 g/100 g) 309 (Table 1B) [8]." Suggestion: "Upon analyzing the eleven non-essential amino acids—Alanine (Ala), Arginine (Arg), Asparagine (Asn), Aspartic acid (Asp), Cysteine (Cys), Glutamic acid (Glu), Glutamine (Gln), Glycine (Gly), Proline (Pro), Serine (Ser), Tyrosine (Tyr)—it emerges that Glu (3.13 g/100 g) is the most abundant amino acid, followed by Arg (2.16 g/100 g) and Asp (1.52 g/100 g) [8]."
"Until now, there were no articles describing all nine essential and all eleven non-essential amino acids together with protein composition in the roasted hazelnut of 314 Italian and Chilean origin."Suggestion: "Until now, there have been no articles describing the composition of all nine essential and all eleven non-essential amino acids together with the protein composition in roasted hazelnuts of Italian and Chilean origin."
"It can be concluded that the total amino acids in the Turkish Tombul hazelnut are 15.10 g/100 g (Table 1C)."Suggestion: "We can conclude that the total amino acids in the Turkish Tombul hazelnut amount to 15.10 g/100 g (Table 1C)."
"It follows that the protein amount this hazelnut variety is 10.42±8.7 (%) [12,13] and in concentration 16.39±1.46 (g/100 g) [8,14]."Suggestion: "It follows that the protein content in this hazelnut variety is 10.42±8.7% [12,13] and its concentration is 16.39±1.46 g/100 g [8,14]."
"Protein concentration in roasted hazelnuts is similar to raw hazelnuts (9.9±8.6 %) (Table 1D)." Suggestion: "The protein concentration in roasted hazelnuts is similar to that in raw hazelnuts, standing at 9.9±8.6% (Table 1D)."
"The carbohydrate amount consisting in simple, complex sugars and fiber was analyzed in the Turkish hazelnut."Suggestion: "The amount of carbohydrates, including simple and complex sugars as well as fiber, was analyzed in the Turkish hazelnut."
"Among simplex sugars Glucose, Fructose, Myo-inositole and Sucrose were revealed, whereas among the complex sugars Raffinose and Stachyose were evaluated."Suggestion: "Among simple sugars, Glucose, Fructose, Myo-inositol, and Sucrose were identified, whereas among the complex sugars, Raffinose and Stachyose were evaluated."
"Surely, is believed that these values are comparable to the Italian hazelnut."Suggestion: "It is believed that these values are comparable to those of the Italian hazelnut."
"However, there is no data regarding carbohydrate concentration in roasted hazelnuts of different origins (Turkey, Italy and Chile)."Suggestion: "However, no data is available regarding the carbohydrate concentration in roasted hazelnuts from different origins (Turkey, Italy, and Chile)."
"In this papers authors firstly extracted lipids from hazelnuts and then, trans-esterified them under alkaline conditions by adding NAPCH3 and methanol as reagents;"Suggestion: "In these papers, the authors first extracted lipids from hazelnuts and then transesterified them under alkaline conditions by adding NAPCH3 and methanol as reagents;"
"A total of twelve different fatty acids among saturated, mono/poly-unsaturated acids were extracted and quantified in the Turkish hazelnut (TABLE 2A);"Suggestion: "A total of twelve different fatty acids, including saturated and mono/poly-unsaturated acids, were extracted and quantified from the Turkish hazelnut (TABLE 2A);"
"Both in Turkish and Italian raw hazelnuts grown in Turkey and Italy respectively, the average of relative % of oleic (C18:1 Δ9), linoleic (C 18:2) and palmitic (C16:0) fatty acids was higher compared to the other acids as indicated in tables 2A and 2B (Average column) [12,13]."
Suggestion: "In both Turkish and Italian raw hazelnuts grown in Turkey and Italy respectively, the average relative percentage of oleic (C18:1 Δ9), linoleic (C 18:2), and palmitic (C16:0) fatty acids was higher compared to the other acids, as indicated in Tables 2A and 2B (Average column) [12,13]."
"Finally, Schlörmann et al., 2015, to discriminate the nutritional benefits of fatty acid-contained Hazelnuts, compared the fatty acids relative of Hazelnut with others different nuts such as Almonds, Macadamia nuts, Pistachios and Walnuts."Correction: "Finally, Schlörmann et al. (2015) aimed to distinguish the nutritional benefits of hazelnuts, which are rich in fatty acids. They compared the relative percentage of fatty acids in hazelnuts with other different nuts, such as almonds, macadamia nuts, pistachios, and walnuts."
"Comparing raw hazelnuts with roasted hazelnuts without peel, it can be seen that the total saturated fatty acids (SFA) are slightly lower in raw hazelnuts and correspond to 7.8 per cent and 8.2 per cent respectively in roasted hazelnuts, monounsaturated to 83.3 per cent and 82.4 per cent and polyunsaturated to 9.3 percent and 9.4 per cent (TABLE 2A)."Correction: "When comparing raw hazelnuts with peeled, roasted hazelnuts, it can be seen that the total saturated fatty acids (SFAs) are slightly lower in raw hazelnuts, corresponding to 7.8 percent in raw and 8.2 percent in roasted hazelnuts. Meanwhile, monounsaturated fats account for 83.3 percent in raw and 82.4 percent in roasted hazelnuts, and polyunsaturated fats account for 9.3 percent in raw and 9.4 percent in roasted hazelnuts (TABLE 2A)."
"Turkish hazelnut (Tombul) contains about 81.2±1.2 % of (C18:1Δ9) oleic fatty acid and about 9.4±1.3 % of C18:2 fatty acid and 5.6±0.3 % of C16:0 palmitic acid."Correction: "The Turkish hazelnut variety, Tombul, contains about 81.2±1.2% of oleic fatty acid (C18:1Δ9), about 9.4±1.3% of C18:2 fatty acid, and 5.6±0.3% of palmitic acid (C16:0)."
"Roasting process does not interfere significantly with FAME [12,13] and thus fatty acids profile, moreover no significant differences were observed among different roasted hazelnut varieties regarding total SFA, MUFA and PUFA analyzed (TABLE 2A-C) "Correction: "The roasting process does not significantly interfere with FAME [12,13], and thus it doesn't affect the fatty acid profile. Moreover, no significant differences were observed among the different roasted hazelnut varieties in terms of total SFA, MUFA, and PUFA analyzed (TABLE 2A-C)."
"palmiticweren’t reduced in the roasted hazelnuts without skin"Correction: "palmitic weren't reduced in the skinless roasted hazelnuts."
"Organic acids or carboxylic acids have a COOH carboxylic group."Suggested correction: "Organic acids, also known as carboxylic acids, have a COOH group."
"In the raw Turkish hazelnut with peel analysed, among these acids the most concentrated is protocatechuic acid (5.31 μg/g) (Table 3A), followed by gallic (2.39 μg/g) and caffeic (1.99 μg/g) acids (TABLE 3A)[12]."Suggested correction: "In the raw Turkish hazelnut with peel that was analyzed, protocatechuic acid (5.31 μg/g) was found to be the most concentrated among these acids, followed by gallic (2.39 μg/g) and caffeic (1.99 μg/g) acids (TABLE 3A)[12]."
"In Shaididi et al. the phenolic acids were obtained from extracts in ethanol, methanol analysed with the Shimadzu HPLC system and the Folin Ciocalteu assay (1965); the concentration is intended to quantify the phenolic acids in μg/g of extract."Suggested correction: "In Shahidi et al., phenolic acids were obtained from ethanol and methanol extracts, which were analyzed with the Shimadzu HPLC system and the Folin Ciocalteu assay (1965); the concentration was measured to quantify the phenolic acids in μg/g of extract."
"The difference in the dominance of phenolic acids in the different studies could probably depend on the extraction/solvent methods used, environmental factors such as harvest time, cultivation and drying methods, season, storage and handling conditions (TABLE 3A) [9]."Suggested correction: "The dominance of different phenolic acids across various studies could probably be attributed to the extraction/solvent methods used, as well as environmental factors such as harvest time, cultivation and drying methods, season, and storage and handling conditions (TABLE 3A) [9]."
"Italian raw hazelnut with skin (TGT, TG) has a higher concentration of protocatechuic acid (5.54 ± 0.33 μg/g), compared to gallic acid (2.18 ± 0.46 μg/g) and caffeic acid 485 (1.88 ± 0.35 μg/g) as observed in Turkish hazelnut [12] and we observed an approximately 25% increase in gallic and caffeic acid concentrations in TGT Piemonte compared to TG Campania (TABLE 3B)."Suggested correction: "The Italian raw hazelnut with skin (TGT, TG) has a higher concentration of protocatechuic acid (5.54 ± 0.33 μg/g), compared to gallic acid (2.18 ± 0.46 μg/g) and caffeic acid (1.88 ± 0.35 μg/g), as observed in Turkish hazelnut [12]. We also observed an approximate 25% increase in gallic and caffeic acid concentrations in TGT Piemonte compared to TG Campania (TABLE 3B)."
Author Response
The revised version is an improvement, but there are still a few areas that could use some refinement for grammar and clarity:
Comments on the Quality of English Language
Answer: We have modified the text according to all the grammatical indications provided to us. The changes in the text are highlighted in yellow.
For example, in the abstract section,
"The objectives of this narrative review will be as follows:" - It's usually best to write in the present tense when talking about the content of the paper, so "The objectives of this narrative review are as follows:" would be better. DONE.
"evaluation of hazelnut bromatological composition" - "Bromatological" is a term that refers to the study of food in terms of its nutritional value and it doesn't need to be hyphenated with a number. This should be "evaluation of the bromatological composition of hazelnuts". DONE.
"raw v/s roasted hazelnut looking for possible nutritional differences among the varieties depending on their production territory, 20 such as Turkey, Italy, and Chile;" - The abbreviation "v/s" isn't standard English. Use "vs." or "versus" instead. The sentence could be more clear if rephrased. Something like: "comparison of the nutritional properties of raw versus roasted hazelnuts, taking into account potential differences among varieties from different production territories such as Turkey, Italy, and Chile;". DONE.
"This review included 27 scientific articles in which the concentrations of macro- and micro-nutrients in hazelnuts subjected to different processing or from different geographical areas or belonging to different varieties around the world, and their concentrations were measured and reported." - The structure of this sentence is a bit confusing. Consider rephrasing to something like: "This review incorporates 27 scientific articles that measured and reported the concentrations of macro- and micro-nutrients in hazelnuts. These hazelnuts were subjected to different processing methods, originated from various geographical areas, or belonged to different varieties." DONE.
"Our results showed that the different varieties and the territory where the hazelnuts is cultivated influence their bromatological composition, and we found that different processing steps can largely influence the concentration of specific nutrients." - The phrase "the hazelnuts is cultivated" should be "the hazelnuts are cultivated" for correct subject-verb agreement. DONE.
"The removal of the skin is particularly critical, which contains a very high concentration of compounds with antioxidant action." - This sentence would be clearer if rephrased to: "The removal of the skin, which contains a very high concentration of compounds with antioxidant action, is particularly critical." DONE.
"Greater attention should be given to the skin by considering it not a waste product, but as an important part of the hazelnut with nutritional properties that are of primary relevance in the Mediterranean diet." - Consider rephrasing for clarity: "We should give greater attention to the skin, considering it not as a waste product, but as an important part of the hazelnut due to its nutritional properties of primary relevance in the Mediterranean diet." DONE.
"A detailed assessment of the nutritional properties of the hazelnut kernel, skin and oil by evaluating the nutrient compositions and possible modifications (increases or reductions) during roasting process, or depending on production territory and origin, has been provided." - This sentence is somewhat confusing. Consider rephrasing to: "We provide a detailed assessment of the nutritional properties of the hazelnut kernel, skin, and oil, evaluating nutrient compositions and possible modifications (increases or reductions) that occur during the roasting process or that depend on the production territory and origin." DONE.
In the introduction section,
"fruits, nuts and hazelnuts" - Hazelnuts are a type of nut, so including "nuts" and "hazelnuts" in the same list could be a bit redundant. You could rephrase this to "fruits, nuts (including hazelnuts)," to make it clear that hazelnuts are part of the broader category of nuts. DONE.
"as flours used to make bread or pasta, as sweets (e.g., nougat, chocolate bars, ice cream, and cakes), or as oil." - It would be clearer to say "in the form of flour used to make bread or pasta, in sweets (e.g., nougat, chocolate bars, ice cream, and cakes), or as oil." DONE.
"In recent decades (since the 1960s) the Mediterranean diet has undergone many changes, with the entry of industrial products and cultural change due to globalization." - This sentence is fine, but it might be clearer to say "Since the 1960s, the Mediterranean diet has undergone many changes due to the introduction of industrial products and cultural shifts caused by globalization." DONE.
"Recent years have seen a further resurgence of nut consumption due to awareness of the beneficial effects of their consumption on health: an intake of 30 g per day of nuts reduces the cardiovascular risk [2]." - This is correct, but you might consider rephrasing for clarity: "Recent years have seen a resurgence in nut consumption due to increased awareness of their health benefits: for instance, consuming 30 g of nuts per day has been linked to reduced cardiovascular risk [2]." DONE.
"In fact, although composed of a very high percentage of fats (60.8 %) compared to most other foods, the content of saturated fatty acids is very low (4.5% in the kernel)." - Rephrase for clarity: "In fact, even though they contain a high percentage of fats (60.8%) compared to most other foods, the content of saturated fatty acids is very low (4.5% in the kernel)." DONE.
"Moreover, monounsaturated fatty acids (MUFA, 45.7% of the seed) and polyunsaturated fatty acids (PUFA, 7.9 % of the seed) are very abundant." - This is fine, but could be rephrased for clarity: "Moreover, they are rich in monounsaturated fatty acids (MUFAs, making up 45.7% of the seed) and polyunsaturated fatty acids (PUFAs, constituting 7.9% of the seed)." DONE.
"In the human gut, phytosterols, due to their high hydrophobicity, interfere with the absorption of cholesterol, helping to lower its concentration in the blood [4]." - Rephrase for clarity: "In the human gut, due to their high hydrophobicity, phytosterols interfere with cholesterol absorption, thus helping to lower its concentration in the blood [4]." DONE.
"Other beneficial effects of consuming nuts, particularly hazelnuts, may result from their high content of L-arginine, which, serving as a substrate for the synthesis of nitric oxide (the main regulator of vascular tone and consequently blood pressure), contributes" - This sentence appears to be incomplete. You could finish it like this: "... contributes to maintaining healthy blood pressure levels." Or, if you have specific information on how L-arginine contributes to health, you could include that instead. DONE.
"Of particular relevance to human health is also the content of micronutrients such as folic acid (vitamin B9) and antioxidants (tocopherols and polyphenols) and their mineral salt composition." - The phrase "is also the content" is awkward. A smoother phrasing might be: "Also of particular relevance to human health is the content of micronutrients such as folic acid (vitamin B9) and antioxidants (tocopherols and polyphenols), as well as their mineral salt composition." DONE.
"Even if all nuts have shared basic nutritional characteristics, however, each has nutritional peculiarities that also vary based on whether or not they have the skin and how they are prepared (raw or toasted)." - The sentence could be smoother without "even if" and "however": "While all nuts share basic nutritional characteristics, each has nutritional peculiarities that vary based on whether they have skin and how they are prepared (raw or toasted)." DONE.
"The hazelnut has the characteristic of being able to be eaten with and without skin and raw or toasted." - This could be simplified to: "Hazelnuts can be eaten with or without skin, and either raw or toasted." DONE.
"It is well known that the hazelnut varieties grown in Turkey are the following with different percentage spread: Tabzon, Giresun, Ordu (40%), Samsun, Akcakoca (20%), Tombul (30%) and other minor varieties." - This sentence could be made clearer: "It is well known that Turkey grows a variety of hazelnuts, including Tabzon, Giresun, Ordu (which make up 40% of the production), Samsun, Akcakoca (20%), Tombul (30%), and other minor varieties." DONE.
"In Italy, on the other hand, we find several varieties that are cultivated the most, in Campania and Lazio followed by Piedmont and then Sicily." - This sentence could be improved: "In Italy, on the other hand, several varieties are predominantly cultivated in Campania and Lazio, followed by Piedmont and Sicily." DONE.
"To date, various research has been published on the nutritional composition of hazelnuts, but there is no review that considers all the nutritional characteristics and the variations of the same related to the roasting and the presence or absence of the skin." - This could be rephrased for clarity: "To date, numerous studies have been published on the nutritional composition of hazelnuts, but no review has comprehensively considered all the nutritional characteristics and how they vary based on factors such as roasting and the presence or absence of skin." DONE.
"Evaluation of Hazelnut bromatological composition and comparison of nutritional properties of raw v/s roasted hazelnut looking for possible nutritional differences among the varieties depending on their production territory, such as Turkey, Italy, and Chile." - Consider rephrasing for clarity: "Evaluation of the bromatological composition of hazelnuts and comparison of the nutritional properties of raw versus roasted hazelnuts, seeking possible nutritional differences among varieties based on their production territory, such as Turkey, Italy, and Chile." DONE.
Remember to maintain the same tense throughout your objectives. As these are objectives for a future review, using the future tense might be more appropriate, e.g., "The objectives of this narrative review will be to evaluate..." etc. This would depend on the overall context of your work.
In the Materials and Methods section,
"A working group was configured as follows: three operators skilled in clinical nutrition, of whom one acting as a methodological operator, two participating as clinical operators; one operator as Molecular Biologist participating as post-doc Researcher for data collection/analysis." - Consider revising for clarity: "A working group was configured as follows: three operators skilled in clinical nutrition, with one acting as a methodological operator, two participating as clinical operators, and one operator, a Molecular Biologist, participating as a post-doc researcher for data collection and analysis." DONE.
"The revision question on the basis of considerations made in the abstract was formulated as follows: “Evaluation of Hazelnut bromatological composition and comparison of nutritional properties of raw v/s roasted hazelnut looking for possible nutritional differences among the varieties depending on their production territory, such as Turkey, Italy, and Chile; Evaluation of nutrients contained in hazelnut skin; Evaluation of nutrients contained in the hazelnut oil.”" - Again, consider revising for clarity: "The review question, based on considerations made in the abstract, was formulated as follows: “Evaluation of the bromatological composition of hazelnuts and comparison of the nutritional properties of raw versus roasted hazelnuts, with the aim of identifying possible nutritional differences among varieties from different production territories, such as Turkey, Italy, and Chile. Additionally, an evaluation of the nutrients contained in hazelnut skin and hazelnut oil." DONE.
"Starting from the data published in 45 scientific papers, including experimental papers and reviews, it was possible to create representative tables of the analysis of the bromatological composition of hazelnuts based on the origin and type of hazelnut processing by including a total of 27 articles in the study." - A slight rephrase for clarity: "Based on the data from 45 scientific papers, including experimental papers and reviews, representative tables were created to analyze the bromatological composition of hazelnuts. These tables considered the origin and type of hazelnut processing, and included data from a total of 27 articles in the study." DONE.
"Different nutrients, including essential and nonessential amino acids, proteins, carbohydrates, fiber, water, fatty and organic acids, phenols, flavonoids, mineral salts, vitamins, antioxidant activity, were evaluated." - Consider revising for clarity: "Various nutrients, including essential and nonessential amino acids, proteins, carbohydrates, fiber, water, fatty and organic acids, phenols, flavonoids, mineral salts, vitamins, and antioxidant activity, were evaluated." DONE.
"Nutrients in the raw hazelnut with skin, roasted without skin/with skin were analyzed, furthermore, the nutritional properties of the skin and finally of the oil derived from the hazelnut were evaluated." - Consider revising for clarity: "The nutrients in raw hazelnuts (with skin), roasted hazelnuts (both with and without skin) were analyzed. Furthermore, the nutritional properties of the hazelnut skin and of the oil derived from hazelnuts were evaluated.” DONE.
In the results section,
"Evaluation of Hazelnut bromatological composition and comparison of nutritional properties of raw v/s roasted hazelnut looking for possible nutritional differences among the varieties depending on their production territory "Suggestion: "Evaluation of hazelnut bromatological composition and comparison of the nutritional properties of raw versus roasted hazelnuts, aiming to identify potential nutritional differences among varieties depending on their production territories." DONE.
"from Turkish grown in Turkey (Tombul)." Suggestion: "from Turkish hazelnuts grown in Turkey (Tombul)." DONE.
"This research was conducted based on the keywords: “hazelnut compositional characteristics” OR “nutritional composition of hazelnut” AND “Impact of roasting” OR “Influence of roasting” and “proteins hazelnut” OR “aminoacids hazelnut”." Suggestion: "This research was conducted using the keywords: “hazelnut compositional characteristics”, “nutritional composition of hazelnut”, “Impact of roasting”, “Influence of roasting”, “proteins in hazelnuts”, and “amino acids in hazelnuts”." DONE.
"from different origin" Suggestion: "from different origins" DONE.
"concentration analysis of mineral salts in the Turkish Tombul raw with skin (R_S) hazelnut and toasted without skin (T_WS) hazelnut."Suggestion: "concentration analysis of mineral salts in the Turkish Tombul raw hazelnut with skin (R_S) and toasted hazelnut without skin (T_WS)." DONE.
"This research was conducted based on the keywords: “skin hazelnut compositional characteristics” OR “nutritional composition of skin hazelnut” AND “skin roasted hazelnut” OR “husk hazelnut” AND “total polyphenol skin hazelnut” AND “phenolic acids skin hazelnut” AND “phenols / flavan-3-ols skin hazelnut” AND “skin hazelnut antioxidant activity”." Suggestion: "This research was conducted using the keywords: “hazelnut skin compositional characteristics”, “nutritional composition of hazelnut skin”, “roasted hazelnut skin”, “hazelnut husk”, “total polyphenols in hazelnut skin”, “phenolic acids in hazelnut skin”, “phenols/flavan-3-ols in hazelnut skin”, and “antioxidant activity of hazelnut skin”." DONE.
"TABLE 7 (A;B;C;D:E) shows the skin analysis of Giresun hazelnut cultivated in Turkey, Tonda Gentile delle Langhe in Chile (TGL_Chile), Tonda Gentile delle Langhe in Italy (TG_Campania) and Tombul in Turkey was added to Table B. " Suggestion: "TABLE 7 (A;B;C;D:E) presents the skin analysis of Giresun hazelnuts cultivated in Turkey, Tonda Gentile delle Langhe in Chile (TGL_Chile), Tonda Gentile delle Langhe in Italy (TG_Campania), and Tombul in Turkey. These findings have been added to Table B." DONE.
"This research was conducted based on the keywords: “oil hazelnut” OR “oil hazelnut compositional characteristics” OR “nutritional composition of oil hazelnut” AND “refined hazelnut oils” AND “oil hazelnut Lipid composition” AND “oxidative stability of oils in hazelnuts” OR “Antioxidative effects on refined hazelnut oil”."Suggestion: "This research was conducted using the keywords: “hazelnut oil”, “hazelnut oil compositional characteristics”, “nutritional composition of hazelnut oil”, “refined hazelnut oils”, “hazelnut oil lipid composition”, “oxidative stability of oils in hazelnuts”, and “antioxidative effects on refined hazelnut oil”." DONE.
"TABLE 8 shows the analysis of hazelnut oil from different origin: Turkish grown in Turkey (Tombul), Italian hazelnut Tonda Giffoni grown in Portugal (TG_Portugal) and in New Zealand (TG_New Zealand)." Suggestion: "TABLE 8 presents the analysis of hazelnut oil from different origins: Turkish hazelnuts grown in Turkey (Tombul), and Italian Tonda Giffoni hazelnuts grown in Portugal (TG_Portugal) and in New Zealand (TG_New Zealand)." DONE.
In the discussion section,
"Evaluation of Hazelnut bromatological composition and comparison of nutritional properties of raw v/s roasted hazelnut looking for possible nutritional differences among the varieties depending on their production territory "Suggestion: "This is an evaluation of the bromatological composition of hazelnuts, comparing the nutritional properties of raw versus roasted hazelnuts. The aim is to identify possible nutritional differences among the varieties depending on their production territories." DONE.
"Few studies currently shed light on the effect of roasting on the bromatological composition of the natural /raw hazelnut and consequently on its nutritional properties."Suggestion: "Currently, there are only a few studies that shed light on the effect of roasting on the bromatological composition of natural/raw hazelnuts and consequently, on its nutritional properties." DONE.
"Thus, it is possible to assess the effects of roasting by evaluating the concentration of micro and macro nutrients such as sugars, fatty acids, organic acids and tocopherols [8]."Suggestion: "Thus, the effects of roasting can be assessed by evaluating the concentrations of micro and macronutrients such as sugars, fatty acids, organic acids, and tocopherols [8]." DONE.
"We included also Locatelli’s papers for our analysis”,Suggestion: "We also included Locatelli’s papers in our analysis. " DONE.
"Analyzing raw natural hazelnut of the Tombul variety (Turkey) Alasalvar et al. show that the most concentrated essential a.a. is Leucine (1.07g/100g), followed by Valine, Phenylalanine and Isoleucine."Suggestion: "In their analysis of raw natural hazelnuts of the Tombul variety from Turkey, Alasalvar et al. found that the most concentrated essential amino acid is Leucine (1.07g/100g), followed by Valine, Phenylalanine, and Isoleucine." DONE.
"Analyzing the eleven non-essential amino acids: Alanine (Ala), Arginine (Arg), Asparagine (Asn), Aspartic acid (Asp), Cysteine (Cys), Glutamic acid (Glu), Glutammin (Gln), Glycine (Gly), Proline (Pro), Serine (Ser), Tyrosine (Try) it emerges that Glu (3.13 308 g/100 g) is the most abundant a.a, followed by Arg (2.16 g/100 g) and Asp (1.52 g/100 g) 309 (Table 1B) [8]." Suggestion: "Upon analyzing the eleven non-essential amino acids—Alanine (Ala), Arginine (Arg), Asparagine (Asn), Aspartic acid (Asp), Cysteine (Cys), Glutamic acid (Glu), Glutamine (Gln), Glycine (Gly), Proline (Pro), Serine (Ser), Tyrosine (Tyr)—it emerges that Glu (3.13 g/100 g) is the most abundant amino acid, followed by Arg (2.16 g/100 g) and Asp (1.52 g/100 g) [8]." DONE.
"Until now, there were no articles describing all nine essential and all eleven non-essential amino acids together with protein composition in the roasted hazelnut of 314 Italian and Chilean origin."Suggestion: "Until now, there have been no articles describing the composition of all nine essential and all eleven non-essential amino acids together with the protein composition in roasted hazelnuts of Italian and Chilean origin." DONE.
"It can be concluded that the total amino acids in the Turkish Tombul hazelnut are 15.10 g/100 g (Table 1C)."Suggestion: "We can conclude that the total amino acids in the Turkish Tombul hazelnut amount to 15.10 g/100 g (Table 1C)." DONE.
"It follows that the protein amount this hazelnut variety is 10.42±8.7 (%) [12,13] and in concentration 16.39±1.46 (g/100 g) [8,14]."Suggestion: "It follows that the protein content in this hazelnut variety is 10.42±8.7% [12,13] and its concentration is 16.39±1.46 g/100 g [8,14]." DONE.
"Protein concentration in roasted hazelnuts is similar to raw hazelnuts (9.9±8.6 %) (Table 1D)." Suggestion: "The protein concentration in roasted hazelnuts is similar to that in raw hazelnuts, standing at 9.9±8.6% (Table 1D)." DONE.
"The carbohydrate amount consisting in simple, complex sugars and fiber was analyzed in the Turkish hazelnut."Suggestion: "The amount of carbohydrates, including simple and complex sugars as well as fiber, was analyzed in the Turkish hazelnut." DONE.
"Among simplex sugars Glucose, Fructose, Myo-inositole and Sucrose were revealed, whereas among the complex sugars Raffinose and Stachyose were evaluated."Suggestion: "Among simple sugars, Glucose, Fructose, Myo-inositol, and Sucrose were identified, whereas among the complex sugars, Raffinose and Stachyose were evaluated." DONE.
"Surely, is believed that these values are comparable to the Italian hazelnut."Suggestion: "It is believed that these values are comparable to those of the Italian hazelnut." DONE.
"However, there is no data regarding carbohydrate concentration in roasted hazelnuts of different origins (Turkey, Italy and Chile)."Suggestion: "However, no data is available regarding the carbohydrate concentration in roasted hazelnuts from different origins (Turkey, Italy, and Chile)." DONE.
"In this papers authors firstly extracted lipids from hazelnuts and then, trans-esterified them under alkaline conditions by adding NAPCH3 and methanol as reagents;"Suggestion: "In these papers, the authors first extracted lipids from hazelnuts and then transesterified them under alkaline conditions by adding NAPCH3 and methanol as reagents;" DONE.
"A total of twelve different fatty acids among saturated, mono/poly-unsaturated acids were extracted and quantified in the Turkish hazelnut (TABLE 2A);"Suggestion: "A total of twelve different fatty acids, including saturated and mono/poly-unsaturated acids, were extracted and quantified from the Turkish hazelnut (TABLE 2A);" DONE.
"Both in Turkish and Italian raw hazelnuts grown in Turkey and Italy respectively, the average of relative % of oleic (C18:1 Δ9), linoleic (C 18:2) and palmitic (C16:0) fatty acids was higher compared to the other acids as indicated in tables 2A and 2B (Average column) [12,13]." Suggestion: "In both Turkish and Italian raw hazelnuts grown in Turkey and Italy respectively, the average relative percentage of oleic (C18:1 Δ9), linoleic (C 18:2), and palmitic (C16:0) fatty acids was higher compared to the other acids, as indicated in Tables 2A and 2B (Average column) [12,13]." DONE.
"Finally, Schlörmann et al., 2015, to discriminate the nutritional benefits of fatty acid-contained Hazelnuts, compared the fatty acids relative of Hazelnut with others different nuts such as Almonds, Macadamia nuts, Pistachios and Walnuts."Correction: "Finally, Schlörmann et al. (2015) aimed to distinguish the nutritional benefits of hazelnuts, which are rich in fatty acids. They compared the relative percentage of fatty acids in hazelnuts with other different nuts, such as almonds, macadamia nuts, pistachios, and walnuts." DONE.
"Comparing raw hazelnuts with roasted hazelnuts without peel, it can be seen that the total saturated fatty acids (SFA) are slightly lower in raw hazelnuts and correspond to 7.8 per cent and 8.2 per cent respectively in roasted hazelnuts, monounsaturated to 83.3 per cent and 82.4 per cent and polyunsaturated to 9.3 percent and 9.4 per cent (TABLE 2A)."Correction: "When comparing raw hazelnuts with peeled, roasted hazelnuts, it can be seen that the total saturated fatty acids (SFAs) are slightly lower in raw hazelnuts, corresponding to 7.8 percent in raw and 8.2 percent in roasted hazelnuts. Meanwhile, monounsaturated fats account for 83.3 percent in raw and 82.4 percent in roasted hazelnuts, and polyunsaturated fats account for 9.3 percent in raw and 9.4 percent in roasted hazelnuts (TABLE 2A)." DONE.
"Turkish hazelnut (Tombul) contains about 81.2±1.2 % of (C18:1Δ9) oleic fatty acid and about 9.4±1.3 % of C18:2 fatty acid and 5.6±0.3 % of C16:0 palmitic acid."Correction: "The Turkish hazelnut variety, Tombul, contains about 81.2±1.2% of oleic fatty acid (C18:1Δ9), about 9.4±1.3% of C18:2 fatty acid, and 5.6±0.3% of palmitic acid (C16:0)." DONE.
"Roasting process does not interfere significantly with FAME [12,13] and thus fatty acids profile, moreover no significant differences were observed among different roasted hazelnut varieties regarding total SFA, MUFA and PUFA analyzed (TABLE 2A-C) "Correction: "The roasting process does not significantly interfere with FAME [12,13], and thus it doesn't affect the fatty acid profile. Moreover, no significant differences were observed among the different roasted hazelnut varieties in terms of total SFA, MUFA, and PUFA analyzed (TABLE 2A-C)." DONE.
"palmiticweren’t reduced in the roasted hazelnuts without skin"Correction: "palmitic weren't reduced in the skinless roasted hazelnuts." DONE.
"Organic acids or carboxylic acids have a COOH carboxylic group."Suggested correction: "Organic acids, also known as carboxylic acids, have a COOH group." DONE.
"In the raw Turkish hazelnut with peel analysed, among these acids the most concentrated is protocatechuic acid (5.31 μg/g) (Table 3A), followed by gallic (2.39 μg/g) and caffeic (1.99 μg/g) acids (TABLE 3A)[12]."Suggested correction: "In the raw Turkish hazelnut with peel that was analyzed, protocatechuic acid (5.31 μg/g) was found to be the most concentrated among these acids, followed by gallic (2.39 μg/g) and caffeic (1.99 μg/g) acids (TABLE 3A)[12]." DONE.
"In Shaididi et al. the phenolic acids were obtained from extracts in ethanol, methanol analysed with the Shimadzu HPLC system and the Folin Ciocalteu assay (1965); the concentration is intended to quantify the phenolic acids in μg/g of extract."Suggested correction: "In Shahidi et al., phenolic acids were obtained from ethanol and methanol extracts, which were analyzed with the Shimadzu HPLC system and the Folin Ciocalteu assay (1965); the concentration was measured to quantify the phenolic acids in μg/g of extract." DONE.
"The difference in the dominance of phenolic acids in the different studies could probably depend on the extraction/solvent methods used, environmental factors such as harvest time, cultivation and drying methods, season, storage and handling conditions (TABLE 3A) [9]."Suggested correction: "The dominance of different phenolic acids across various studies could probably be attributed to the extraction/solvent methods used, as well as environmental factors such as harvest time, cultivation and drying methods, season, and storage and handling conditions (TABLE 3A) [9]." DONE.
"Italian raw hazelnut with skin (TGT, TG) has a higher concentration of protocatechuic acid (5.54 ± 0.33 μg/g), compared to gallic acid (2.18 ± 0.46 μg/g) and caffeic acid 485 (1.88 ± 0.35 μg/g) as observed in Turkish hazelnut [12] and we observed an approximately 25% increase in gallic and caffeic acid concentrations in TGT Piemonte compared to TG Campania (TABLE 3B)."Suggested correction: "The Italian raw hazelnut with skin (TGT, TG) has a higher concentration of protocatechuic acid (5.54 ± 0.33 μg/g), compared to gallic acid (2.18 ± 0.46 μg/g) and caffeic acid (1.88 ± 0.35 μg/g), as observed in Turkish hazelnut [12]. We also observed an approximate 25% increase in gallic and caffeic acid concentrations in TGT Piemonte compared to TG Campania (TABLE 3B)." DONE.